# An integrated probabilistic assessment to analyze stochasticity of soil erosion in different restoration vegetation

Ji Zhou [1,2,3], Bojie Fu [1,2,]* Guangyao Gao [1,2], Yihe Lü [1,2], Shuai Wang [1,2]

[1] State Key Laboratory of Urban and Regional Ecology, Research Center for Eco-Environmental Science, Chinese Academy of Science, Beijing 100085, PR China,

[2] Joint Center for Global Change Studies, Beijing 100875, PR China

[3] University of Chinese Academy of Sciences, Beijing 100049, PR China

*Corresponding author: Bojie Fu

E-mail: bfu@rcees.ac.cn

Address: State Key Laboratory of Urban and Regional Ecology, Research Center for Eco-Environmental Science, Chinese Academy of Science, P. O. Box 2871, Beijing 100085, PR China

**Abstract**:

Stochasticity of soil erosion reflects the variability of soil hydrological response to precipitation under complex environment. Assessing this stochasticity is important to conserve soil and water resources, however stochasticity of erosion event in restoration vegetation types in water-limited environment is less investigated. In this study, we constructed an event-driven framework to quantify the stochasticity of runoff and sediment generation in three typical restoration vegetation types (*Armeniaca sibirica* (T1), *Spiraea pubescens* (T2), and *Artemisia copria* (T3)) at closed runoff plot over five rainy seasons in the Loess Plateau of China. The results indicated that, under the same rainfall condition, the average probabilities of runoff and sediment in T1 (3.8% and 1.6%) and T3 (5.6% and 4.4%) were lowest and highest, respectively. The Binomial and Poisson probabilistic model were two effective ways to simulate the frequencies distribution of times of erosion events occurring in all restoration vegetation. The Bayes model indicated that relative longer duration and stronger intensity rainfall events respectively become the main probabilistic contributors of one stochastic erosion event occurring in T1 and T3. Logistic regression modeling highlighted that the higher-grade rainfall intensity and canopy structure were as two most important factors to respectively improve and restrain the probability of stochastic erosion generation in all restoration vegetation types. Bayes, Binomial, Poisson and logistic regression models constituted an integrated probabilistic assessment to systematically simulate and evaluate soil erosion stochasticity. It may be an innovative and important complement in understanding of soil erosion from stochasticity view, and also provide an alternative

to assess the efficacy of ecological restoration on conserving soil and water resource in
a semi-arid environment.
**Key words**: stochasticity of soil erosion, Binomial and Poisson, logistic regression
model, restoration vegetation,





**1.  Introduction**

Soil erosion is one of globe environmental problems. In the recent centuries, the erosion rate over worldwide has been accelerating by the climate change and anthropogenic activities, causing soil deterioration and terrestrial ecosystem degradation (Jiao et al., 1999; Marques et al., 2008; Fu et al., 2011; Portenga and Bierman, 2011). The uncertainty and intensity of soil erosion constitute the main feature of erosive phenomenon. Although many studies have been concentrating on the intensity of erosion under different spatiotemporal scales (Cantón et al., 2011; Puigdefábregas et al., 1999), the uncertainty of soil erosion generation is another challenge of researchers expecting to improve the accuracy of erosion prediction. To some extent, the stochasticity of environment and spatiotemporal heterogeneity of soil loss mainly affected the randomness of runoff production and sediment transportation in natural conditions (Kim. J et al., 2016). Meanwhile, the complex mechanism of erosion generation also increased the uncertainty and variation of erosion processes (Sidorchuk, 2005, 2009). Therefore, how to effectively describe the erosive stochasticity and to reasonably assess its impacting factors is necessary and important for understating soil erosion science from the perspective of randomness.

First, the combination of various probabilistic, conceptual and physical models have been reported as different integrated approaches to describe the stochasticity of soil erosion intensity (Table 1). As one form of erosion intensity, the runoff processes was proved as a stochastic process by different mathematic simulation models. Some studies (Moore, 2007; Janzen and McDonnell, 2015) have also simulated the stochastic

processes, and further quantified the randomness of runoff production and its
connectivity dynamics in hillslope and catchment scales by using different probabilistic
distribution functions and conceptual models. In these studies, the theory-driven
conceptual models simplified main hydrological behaviors related to runoff production,
highlighting the stochastic effects of infiltration and precipitation on runoff processes.
Based on above precondition, the data-driven probabilistic models further simulated the
stochastic runoff production by mapping or calibrating the difference between observed
and predicted probabilistic values. As a results, the stochastic-conceptual approaches
have formed an effective framework to model the rainfall-runoff processes (Freeze,
1980), as well as to assess flood forecasting (Yazdi et al., 2013)
The stochasticity of soil erosion rate which is another pattern of erosion intensity was
generally investigated by probabilistic and physical models in some studies. The
theory-driven physical models in these studies (Sidorchuk, 2005) integrated
hydrological and mechanical mechanism of overflow and soil structure with sediment
transpiration processes, stressing the stochastic effect of physical principles on erosion
rate in different spatial scales (Table 1). Especially Sidorchuk in 2005 further introduced
stochastic variables and parameters into probabilistic models by randomizing the
physical properties of overflow and soil structure. This approach developed the
understanding of uncertainty of sediment transpiration processes, leading the
randomness simulation to be better fit the reality of stochastic erosion rate (Sidorchuk,
2009). Additionally, the stochasticity of soil erosion rate also reflected the erosion risk
which was assessed by the integration of theory-driven empirical model with
geostatistics (Jiang et al., 2012; Wang et al., 2002; Kim. J et al., 2016). Erosion risk
analysis generally concentrated on the uncertainty or variability of soil erosion rate in
catchment and regional scales. It highlighted the impact of the spatiotemporal
heterogeneous rainfall and other environment conditions on the stochastic erosion rate.
In a word, these probabilistic and physical models constituted a systematical analysis
framework which closely related to the principle of water balance and basic
hydrological assumptions. It effectively described the randomness of soil erosion rate
affected by complex hydrological processes (Bhunya et al., 2007). However, few
studies has been made to analyze the stochasticity of soil erosion events. Especially,
there are little effort to systematically investigate how the signal of stochastic rainfall
is transmitted to erosion event occurring in different restoration vegetation types based
on observational data rather than on other model assumptions. In fact, this event-based
investigation deriving from specific experiment results probably have more practical
meaning for understanding the stochastic interaction between rainfall and erosion
events.
Secondly, the probabilistic approaches have also been reported as a crucial tool to
describe the stochasticity of factors affecting soil erosion rate (Table 1). The
randomness of soil water content (Ridolfi et al., 2003), antecedent soil moisture
(Castillo et al., 2003), infiltration rate (Wang, P and Tartakovsky 2011) and soil
erodibility (Wang et al., 2001) in heterogeneous soil types were all modelled by
different probability distribution functions. These stochasticity of soil hydrological
characteristics related to erosion rate mainly acted as various roles on impacting the
spatiotemporal distribution of erosion rate especially generating in regional or even
larger spatial scales. Meanwhile, as the main driving force of soil erosion generation,
the uncertainty of rainfall event, to some extent, represents the environment
stochasticity (Andres-Domenech et al., 2010). Eagleson in 1978 applied probabilistic-
trait models to characterize the stochasticity of rainfall event by using Poisson and
Gamma probability distribution functions. The stochastic rainfall distribution in
different spatiotemporal scales has also been applied to examine the effect of runoff and
sediment yield (Lopes, 1996), to calibrate the runoff-flood hydrological model
(Haberlandt and Radtke, 2014), as well as to evaluate the sewer overflow in urban
catchment (Andres-Domenech et al., 2010).
It has been well recognized the role of spatial distribution of vegetation in controlling
the soil erosion rate under different spatiotemporal scales (Wischmeier and Smith, 1978;
Puigdefabregas, 2005). How the plants reduce soil erosion rate was also illuminated
and interpreted by various physical and empirical models (Liu, 2001; Mallick et al.,
2014; Prasannakumar et al., 2011). In theory, Puigdefabregas in 2005 proposed
Vegetation-Driven-Spatial-Heterogeneity (VDSH) to explain the relationship between
vegetation patterns and erosion fluxes, which improves the understanding of
hydrological function of plant on erosion processes. Moreover, Trigger-Transfer-
Reserve-Pulse (TTRP) framework proposed by Ludwig in 2005, systematically
explored the responses and feedback between vegetation patches and runoff-erosion
during whole ecohydrological processes. Theoretically, the stochastic signals of
different rainfall events could also be disturbed by the hydrological function of plant,
which finally affects the randomness of runoff and sediment events occurring in various
vegetation types. However, little effort has been made to explore the effect of different
vegetation types on the stochasticity of corresponding soil erosion events. In particular,
less approaches have been used to analyze how the properties of rainfall, soil and
vegetation impact on the stochastic erosion events through event-based investigation
deriving from observational data rather than on theory-based models. Actually, logistic
regression modeling (LRM) probably deal with above problems. LRM evaluates the
causal effects of categorical variables on dependent variables, and quantifies the
probabilistic contribution of influencing factors on the randomness of responsive
random events in terms of odds ratio (Hosmer et al., 2013). It could be regarded as
another probabilistic model to explore the probability-attribution of influencing factors.
However, little literature is available on making LRM to explore the probabilistic
attributing of stochastic erosion events under complex environmental conditions.
In this study, we hypothesized that the uncertainty of erosive events was also an
important property of soil erosion phenomenon, and monitored erosion events
generating in three typical restoration vegetation types in runoff plots scale over
consecutive five years' rainy seasons. We aim to (1) comprehensively describe the
stochasticity of runoff and sediment events in details by using probability theory, and
(2) to systematically evaluate the effect of the properties of soil, plant and rainfall on
the stochastic erosion events by employing LRM approach. The probabilistic
description-attribution approach could constitute an integrated probabilistic assessment
based on event-driven probability theory and data-drive experimental observation.
Meanwhile, the investigation of stochastic soil erosion events by the integrated
assessment may be an innovative and important complement in understanding of soil
erosion from stochasticity view, but also could provide an alternative to assess the
efficacy of ecological restoration on conserving soil and water resource in a semi-arid
environment.

183                                         Table 1


**2.  Method**
**2.1 Definition and classification of random events**
Each observed stochastic weather condition with different durations in field monitoring
period was defined as a random experiment. All possible outcomes of a random
experiment constituted a sample space ($\Omega$) defined as a random observational event
(short for O event). Two mutually exclusive random event types—random rainfall event
(short for I event) and random non-rainfall event (short for C event)—constituted the O
event. Precipitation is a necessary condition of runoff generation, and the random runoff
production event (short for R event) is a subset of I event. Similarly, R event is also a
necessary condition of random sediment migration event (short for S event), which lead
to S event be a subset of R event. As a result, O, C, I, R, and S events constituted a
random events framework (OCIRS) to reflect the stochasticity of environment in which
soil erosion events generation.

198       The random event duration in OCIRS is an important property of stochastic weather

conditions. In particular, the duration property of I event was closely related to the
transmission of stochastic signals of rainfall into the R and S events. According to the
rainfall duration patterns in China (Wei et al., 2007), the time interval between two
adjacent individual I events is set to be more than 6 hours, forming the criteria for
individual rainfall classification. Meanwhile, based on the observation of random
events over five consecutive rainy seasons, we summarized duration property of all I
events and further classified them into four mutually exclusive I event types. They were
a random extreme long rainfall event type (short for Ie event), a random general long
duration rainfall event type (short for Il event), a random spanning rainfall event type
(short for Is event) whose duration spans two consecutive days, and a random within
rainfall event type (short for Iw event) generated in a day. Additionally, the C event can
also be divided into two types at daily scale. They are random non-rainfall event type
lasting a whole day (short for Cd event), and random non-rainfall event type whose
duration is less than 24 hours (short for Ch event) which is interrupted by I event.
Table 2 indicated the physical, probabilistic properties and implications of all random
event types in OCIRS. The classification process of all random event types was
sketched by figure 1a, the Venn diagram of all random event types in OCIRS was
showed in figure 1c. Considering the observed longest duration of Ie event
approximating 72 hours, in figure 1b, we summarized a series of random event
sequences in terms of different combing patterns of I and C events in every three
consecutive days during the whole monitoring period.

221                                        Figure 1

222                                         Table 2



**2.2 Probabilistic description of erosion event**
**2.2.1 Conditional probability of erosion event**
In the sample space $\Omega$, for any random event type $E$ in OCIRS, we defined $P(E)$ as the
proportion of time that $E$ occurs in terms of relative frequency:
$$P(E) = \lim_{n \to \infty} \frac{n(E)}{n} = p_E, \ \ p_E \in [0,1] \tag{1}$$
Theoretically, $n(E)$ is the number of times in $n$ outcomes of observed random
experiment that the event $E$ occurs. According to the law of total probability (Robert et
al., 2013), the probability of R event is defined as:
$$P(R) = P(RI) = P(R| \cup_{m=1}^{4} I_m)P(\cup_{m=1}^{4} I_m) = \sum_{m=1}^{4} P(R|I_m)P(I_m) = p_R \tag{2}$$
$I_m$, m=1, 2, 3 and 4 represent the Ie, Il, Is, and Iw respectively, and $P(R|I_m)$ represents
conditional probability that R event occur given that $m^{th}$ I event type has occurred.
Similarly, the probability of S event is defined as:
$$P(S) = P(SI) = P(S| \cup_{m=1}^{4} I_m)P(\cup_{m=1}^{4} I_m) = \sum_{m=1}^{4} P(S|I_m)P(I_m) = p_S \tag{3}$$
Equation (2) and (3) quantify the stochastic soil erosion events by using conditional
probability.
**2.2.2 Probability distribution functions of erosion event**
We defined X, Y as two discrete random variables, representing two real-valued
functions defined on the sample space ($\Omega$). Let X, Y denote the numbers of times of R
and S events occurrence respectively, and assign the sample space $\Omega$ to another random
variable Z. $X(R) = x, Y(S) = y, Z(\Omega) = z, y \leq x \leq z$. $x$, $y$, $z$ are integers. The ranges
of X and Y are $R_X = \{all\ x: x = X(R), all\ R \in \Omega\}$ and $R_Y = \{all\ y: y =$
$Y(S), all\ S \in \Omega\}$. The probability of $x_i$ or $y_j$ numbers of times of R or S events can
be quantified by probability mass function (PMF) as follow:
$$pmf_X(x_i) = P[\{R_i: X(R_i) = x_i,\ x_i \in R_X\}] \tag{4}$$
$$pmf_Y(y_j) = P[\{S_j: Y(S_j) = y_j,\ y_j \in R_Y\}]\ \text{for}\ i \geq j \tag{5}$$
PMF in Equation (4), (5) describe the general expression of probability distribution of
all possible numbers of times of R or S events.
The random variables X, Y obey the Binominal distribution with $n$ independent
Bernoulli experiments (Robert et al., 2013). Therefore, the PMF of X, and Y can be
defined as follow:
$$pmf_{Xbin}(x) = P_{Xbin}(X = x) = \begin{cases} \binom{n}{x} p_R^x (1 - p_R)^{n-x} & x = 0,1,2,\dots,n \\ 0 & elsewhere \end{cases} \tag{6}$$
$$pmf_{Ybin}(y) = P_{ybin}(Y = y) = \begin{cases} \binom{n}{y} p_S^y (1 - p_S)^{n-y} & y = 0,1,2,\dots,n \\ 0 & elsewhere \end{cases} \tag{7}$$
where $x$ and $y$ indicate all possible numbers of times of R and S occurring over $n$ I
events. However, when the Bernoulli experiment is performed infinite independent
times ($n\rightarrow\infty$), the Binomial PMF can be transformed into Poisson PMF (proved by
appendix A), and finally expressed as follow:
$$pmf_{Xpoi}(x) = P_{Xpoi}(X = x) = \begin{cases} \dfrac{\lambda_R^x e^{-\lambda_R}}{x!} & x = 0,1,2,\dots \\ 0 & elsewhere \end{cases} \tag{8}$$
$$pmf_{Ypoi}(y) = P_{Ypoi}(Y = y) = \begin{cases} \dfrac{\lambda_S^y e^{-\lambda_S}}{y!} & y = 0,1,2,\dots \\ 0 & elsewhere \end{cases} \tag{9}$$
where the parameter $\lambda_R \approx np_R, \lambda_S \approx np_S$. Equation (6) ~ (9) reflect two PMF models
to simulate the probability distribution of R or S events.

**2.3 Probabilistic attribution of erosion events**

**2.3.1 Bayes model**

Based on the Bayes forumula theroy (Sheldon, 2014), if we want to evaluate how much
the probabilistic contributions of $k^{th}$ type of random rainfall event on one stochastic
runoff or sediment event which has been generated and observed, the Bayes model can
quantify the results as follow:

$$P(I_k|R) = \frac{P(I_kR)}{P(R)} = \frac{P(R|I_k)P(I_k)}{\sum_{m=1}^{4} P(R|I_m)P(I_m)} \tag{10}$$

$$P(I_k|S) = \frac{P(I_kS)}{P(S)} = \frac{P(S|I_k)P(I_k)}{\sum_{m=1}^{4} P(S|I_m)P(I_m)} \tag{11}$$

In fact, the Bayes model provides an important explanation that how the priori
stochastic information $(P(I_k))$ was modified by the posterior stochastic information
$(P(R) \text{or} P(S))$. The application of Bayes model in equation (10) ~ (11) reflects the
feedback of random erosion events on the stochastic rainfall events. It could also be
regarded as one pattern of probabilistic attribution to assess the effect of different
random rainfall events on the uncertainty of soil erosion events without considering the
diversity of restoration vegetation.

**2.3.2 Logistic regression model**

Firstly, we constructed event-driven logistic function, and defined $Y_R$ and $Y_S$ as two
dichotomous dependent variables. When we denoted 1 or 0 to $Y_R$ and $Y_S$ respectively, it
means that a R and S event has occurred or not occurred. Given $Y_R$ is a dichotomous
dependent variable of R event in linear probability model to be expressed as follow:

$$Y_{R_i} = \alpha + \beta_1 x_{1i} + \beta_2 x_{2i} + \cdots + \beta_n x_{ni} + \xi_i = \alpha + \sum_{n=1}^{n} \beta_n x_{ni} + \xi_i \tag{12}$$

Then further transforming equation (12) into conditional probability of R event which

has generated in $i^{\text{th}}$ observation time as follow:

$$P\left(Y_{R_i} = 1 \middle| \cap_{n=1}^{n} x_{ni}\right) = P\left[\left(\alpha + \sum_{n=1}^{n} \beta_n x_{ni} + \xi_i\right) \geq 0\right]$$

$$= P\left[\xi_i \leq \left(\alpha + \sum_{n=1}^{n} \beta_n x_{ni}\right)\right]$$

$$= F\left(\alpha + \sum_{n=1}^{n} \beta_n x_{ni}\right) \tag{13}$$

$\alpha, \beta$ are constants, $F(\alpha + \sum_{n=1}^{n} \beta_n x_{ni})$ is the cumulative distribution function of $\xi_i$

when $\xi_i = \alpha + \sum_{n=1}^{n} \beta_n x_{ni}$. Equation (12) and (13) quantified the stochasticity of $Y_{R_i}$

depending on the linear combination of $n$ influencing factors $x_n$ and measurement error

$\xi$ under $i^{\text{th}}$ observation times of stochastic runoff generation.

Secondly, assuming the probabilistic distribution of $\xi_i$ satisfies logistic distribution

and $P\left(Y_{R_i} = 1 \middle| \cap_{n=1}^{n} x_{ni}\right) = p_i$, then the logistic regression modeling (LRM)

expression of $Y_{R_i} = 1$ is deduced as follow:

$$p_i = F\left(\alpha + \sum_{n=1}^{n} \beta_n x_{ni}\right) = \frac{1}{1 + e^{-(\alpha + \sum_{n=1}^{n} \beta_n x_{ni})}} = \frac{e^{\alpha + \sum_{n=1}^{n} \beta_n x_{ni}}}{1 + e^{\alpha + \sum_{n=1}^{n} \beta_n x_{ni}}} \tag{14}$$

Correspondingly, the LRM of $Y_{R_i} = 0$ can be express as:

$$P\left(Y_{R_i} = 0 \middle| \cap_{n=1}^{n} x_{ni}\right) = 1 - p_i = \frac{1}{1 + e^{\alpha + \sum_{n=1}^{n} \beta_n x_{ni}}} \tag{15}$$

The ratios of equation (14) to (15) is defined as odds of R event:

$$\text{Odds} = \frac{p_i}{1 - p_i} = \frac{\dfrac{e^{\alpha + \sum_{n=1}^{n} \beta_n x_{ni}}}{1 + e^{\alpha + \sum_{n=1}^{n} \beta_n x_{ni}}}}{\dfrac{1}{1 + e^{\alpha + \sum_{n=1}^{n} \beta_n x_{ni}}}} = e^{\alpha + \sum_{n=1}^{n} \beta_n x_{ni}}, \ \text{odds} \in [0, 1] \tag{16}$$

In this study, the odds in equation (16) is a probabilistic attribution index to quantify

how much the $n$ influencing factors to affect the generation of $i^{\text{th}}$ stochastic runoff event.

Specifically, when the odds of an influencing factor is greater (less) than 1, it means

that the corresponding influencing factor exerts positively (negatively) effects on the

probability of R generation.

Finally, taking the natural logarithms of the both sides of equation (16), we transform

the odds of stochastic runoff event into linear equation (17) reflecting the standard

expression of LRM:

$$ln\left[\frac{P\left(Y_{R_i}=1\middle|\cap_{n=1}^{n}x_{ni}\right)}{P\left(Y_{R_i}=0\middle|\cap_{n=1}^{n}x_{ni}\right)}\right]=ln\left(\frac{p_i}{1-p_i}\right)=\alpha+\sum_{n=1}^{n}\beta_n x_{ni} \qquad (17)$$

LRM could be regarded as another probabilistic attribution pattern to evaluate the effect

of mutiple impacting factors—such as soil, vegetation, and rainfall—on the randomness

of soil erosion events occuring in different restoration vegetation types.

**3.  Experimental design and data analysis**

**3.1 Study area**

The study was implemented in the Yangjuangou Catchment (36 ˚42'N, 109 ˚31'E, 2.02

km$^2$) which is located in the typical hilly-gully region of the Loess Plateau in China

(Figure 2a). A semi-arid climate in this area is mainly affected by the North China

monsoon. Annual average precipitation reaches approximately 533 mm, and the rainy

season here spans from June to September (Liu et al., 2012). The Calcaric Cambisol

soil type (FAO-UNESCO, 1974) with weak structure and higher erodibility in the Loess

Plateau is vulnerable to water erosion. For these reasons, soil and water loss was one of

most environmental problems to seriously degrade the ecosystem in the Loess Plateau

before 1980s (Miao et al., 2010; Wang et al., 2015). After that, as a crucial soil and

water resource protection project, the Grain-for-Green Project was widely implemented

in the Loess Plateau. A large number of steeply sloped croplands were abandoned,
restored or natural recovered by local shrub and herbaceous plants (Cao et al., 2009;
Jiao et al., 1999). In the Yangjuangou Catchment, the main restoration vegetation
distributed on hillslopes includes *Robinia. pseudoacacia Linn*, *Lespedeza davurica,*
*Aspicilia fruticosa, Armeniaca sibirica, Spiraea pubescens*, and *Artemisia copria*, etc.
All the restoration vegetation was planted over 20 years ago.
**3.2 Design and monitoring**
In the Yangjuangou Catchment, we have had conducted a systematic long-term field
experiments, including the monitoring of soil erosion (Liu et al., 2012; Zhou et al.,
2016), observation of soil moisture dynamic (Wang et al., 2013; Zhou et al., 2015) and
assessment of soil controlling service in this typical water-restricted environment (Fu
et al., 2011).
In this study, we first monitored the soil erosion events occurring in three typical
restoration vegetation (*Armeniaca sibirica* (T1), *Spiraea pubescens* (T2) and *Artemisia*
*copria* (T3)) from rainy season of 2008 to 2012 (figure 2b). Each restoration vegetation
type was designed in three 3 m by 10 m closed runoff-plot distributing on southwest
facing hillslopes with a 26.8% aspect. The boundaries of each runoff-plot were
perpendicularly fenced by impervious polyvinylchloride (PVC) sheet with 50 cm depth.
Collection troughs and storage buckets were installed at the bottom boundary to collect
the runoff and sediment (Zhou et al., 2016). Under natural precipitation condition, we
recorded the number of times of stochastic runoff and sediment events generating in
each runoff-plot over five rainy seasons. Meanwhile, we collected runoff and sediment,
and separated them after settling the collecting bottles for 24 hours, dried at 105℃ over
8 hours and weighted.
Secondly, we systematically monitored the hydrological properties of soil in different
restoration vegetation types. In the rainy season of 2010, the dynamic of soil moisture
was started to be measured in the study region (Wang et al., 2013). The real-time
dynamic data of soil water content with interval of 10 minutes were recorded by the S-
SMC-M005 soil moisture probes (Decagon Devices Inc., Pullman, WA), and were
collected by HOBO weather station logger (figure 2c). These data provided the
information about average antecedent soil moisture (short for ASM) before every
rainfall events generating in the two rainy seasons from 2010 to 2012. We further
measured the field saturated hydraulic conductivity (short for SHC) in all vegetation
types by Model 2800 K1 Guelph Permeameter (Soilmoisture Equipment Corp,. Santa
Barbara, CA, USA) to determine the average infiltration capability of soil matrix (figure
2d).
Thirdly, we also investigated the morphological properties of different vegetation
types in each runoff-plot for 2-3 times over different periods of rainy season. We
measured the average crown width, height and the thickness of litter layer in three
restoration vegetation by setting $60 \times 60$ cm quadrats in each runoff plot (Bonham, 1989)
(figure 2e).
Finally, two tipping bucket rain gauges were installed outside of runoff-plot to
automatically record the rainfall processes over the five rainy seasons with an accuracy
of 0.2 mm precipitation. Table 3 summarized the properties of four types of random
rainfall event, and all the basic characteristic of soil and vegetation was showed in Table

373  4.

374                                 Figure 2

375                                 Table 3

376                                 Table 4


**3.3 Statistics**
We employed nonparametric statistical tests—one-way ANOVA and post hoc LSD—to
determine the significant difference of soil, vegetation and erosive properties in the
three restoration vegetation types. meanwhile, the maximum likelihood estimator (MLE)
and uniformly minimum variance unbiased estimator (UMVUE) (Robert et al., 2013)
were explored to compare the suitability of the binomial PMF and Poisson PMF for
predicting the uncertainty of runoff and sediment generation over long term.

**4.  Results**
**4.1 Environmental stochasticity in different rainy seasons**
The probabilistic distribution of random rainfall events (I events) and random non-
rainfall events (C events) forms the environmental stochasticity which is a background
of stochastic soil erosion generation. In the OCIRS, the stochastic environment at
monthly and seasonal scales over five rainy seasons was described by figure 3. From
the rainy season of 2008 to 2012, the probability of I event generation firstly increased
with the increasing of monitoring period and then decreased in the last two rainy
seasons. In the rainy season of 2008, the average probability of I event was lower than
other four rainy seasons, with being less than 15%. However, the types of I events was
most complex in 2008. The random extreme long rainfall event (Ie event) only appeared
in this rainy season, with the probability even reaching to 2.5% On the other hand, the
average probability of I event was the highest in the rainy season of 2010, with being
larger than 18%. But, there only existed two types of I events (Iw and Is events) in this
rainy season. Over the five rainy seasons, the average probability of Iw (12.3%) and Ie
(0.8%) events generation were the highest and lowest, respectively. The average
probability of Is (1.7%) and Il (1.3%) events ranged between Iw and Ie. The probability
of Cd event was higher than Ch in each month of rainy season, with average probability
being 54.4% and 29.4%, respectively. Moreover, in the table 3, the difference of average
precipitation and duration in the four types of I events was significance. But the average
rainfall intensity of Iw and Is events were nearly twice that of Il and Ie events.

408                                                     Figure 3


**4.2 Stochasticity of soil erosion events**
**4.2.1 Probability of erosion events in vegetation types**
The stochasticity of erosion events was quantified by the probability of runoff and
sediment generation in three restoration vegetation types (T1, T2 and T3) under
monthly and rainy season scales (figure 4). Over the five rainy seasons, the probability
of soil erosion occurring in all vegetation types generally decreased with the increasing
of monitoring period, and then increased in 2012. At early period of erosion monitoring
(2008), the randomness of erosion events is similar, and the probability of R and S event
ranged from 6% to 13% and from 3% to 13% respectively. After that, from rainy season
of 2009 to 2011, the highest probabilities of erosion events in each vegetation type all
concentrated in the July and August of each season. As to runoff production, the average
probability of R event in T1 (3.78%) was less than that of T2 (5.60%) and T3 (5.58%)
under same precipitation condition. With respect to sediment yield, the average
probability of S event in T1 (1.65%) was also the lowest in all restoration vegetation
types. Especially, in the last two rainy seasons, there was no S event occurring in T1,
but, the average probability of S event in T2 and T3 reached to 1.83% and 3.36%
respectively in corresponding rainy seasons. Consequently, affected by the same
stochastic signal of rainfall events, T1 and T3 have the lowest and highest probability
of erosion event generation over the five rainy seasons respectively.

430                                    Figure 4


**4.2.2 Probabilistic distribution of erosion events in vegetation types**
More detailed stochastic information of erosion events in different vegetation types was
simulated by Binominal and Poisson mass functions (PMFs) under the monthly scales.
It also compared the frequencies distribution of different numbers of observed erosion
events with the corresponding simulated results by the two PMFs in figure 5. Firstly, as
to the detailed stochastic information of R events, the two PMFs generally provided a

better simulation to the observation in T1 than that of in T2 and T3. When comparing

the simulated and observed values, Binomial PMF supplied better simulation to the

observed numbers of time of R events with larger frequency (such as 2~4 time) than

that of Poisson PMF. However, Poisson PMF simulated the observed numbers of time

of R events with the lower frequency (such as 6~8 times) better than that of Binomial

PMFs. Secondly, as to the detailed stochastic information of S event, the two PMFs

provided better simulation to the observation in T3 than that of in T1 and T2. In

particular, when the number of times of S event generation reaches two in T1 and T2,

the corresponding simulated probability values were all nearly 2 times larger than the

observed frequencies, reflecting the most simulation error of the two PMFs. Moreover,

with the restoration vegetation types changing from T1 to T3, both of the simulated and

observed numbers of time of R and S events with largest probability or frequency

increased in consistence. In a word, comparing with the observed frequency of numbers

of erosion events, both PMFs indicated well-simulating effect to detail the stochasticity

of runoff and sediment events under monthly scale.

Figure 5

**4.3 Stochastic attribution of soil erosion events**

**4.3.1 Effect evaluation of stochastic erosion events by Bayes model**

The Bayes model was applied to analyze the effect of random rainfall events (including

Iw, Is, Il and Ie) on stochastic erosion events in different restoration vegetation types.

Specifically, Bayes model evaluated the different probabilistic contributions of four
types of I events on one observed erosion event which has stochastically generated in
specific vegetation type under monthly and rainy seasonal scales (figure 6). In the rainy
season of 2008, the types of I events driving one stochastic erosion event was most
complex than other rainy seasons. In contrast, one stochastic soil erosion generation in
three vegetation types attributed to only Iw and Is events in the rainy season of 2010.
In other three rainy seasons, when one R or S event stochastically generated on T1, the
main contributing I event types concentrated on Is and Il events both of which have
relatively higher precipitation and longer duration, respectively. On the other hand, if
one R or S event occurred in T2 or T3 randomly, the main contributing I event types
was Iw event which, however, have no contribution to S event occurred on T1.

471        In general, over five rainy seasons, the composition of I event driving one R event

was more complex than that of driving one S event. The relative longer duration rainfall
events (Il and Ie) became the main probabilistic contributors of one stochastic erosion
event occurring in T1, and the relative stronger intensity rainfall events (Iw and Is)
mainly caused one random erosion event generating in T2 and T3.

477                                     Figure 6


**4.3.2 Effect evaluation of stochastic erosion events by LRM**
According to the results of significant difference analysis in table 4, we defined the
properties of soil and plant as ordinal variables, and classified them into four grades
(Table 5). Meanwhile, based on previous studies (Liu et al., 2012; Wei et al., 2007) and
rainfall properties in this study area, we further subdivided all precipitation and rainfall
intensity into four grades with different scores.
First, the intensity of positive and negative effects of single influencing factor on the
probability of runoff and sediment generation in all restoration vegetation types was
quantified in terms of odds ratio of erosion events by LRM (table 6). In the LRM, the
highest and lowest odd ration appeared in rainfall intensity ordinal variable (INT) and
average crown width ordinal variable (CRO). The increasing INT and CRO (from
middle to extreme grade) significantly increased and decreased the odds ratio of erosion
events, respectively. This means that INT and CRO acts as two most important roles on
improving and restraining the probability of stochastic erosion generation in all
restoration vegetation types. Additionally, the increasing of antecedent soil moisture
ordinal variable (ASM) and the filed saturated hydraulic conductivity ordinal variable
(SHC) (from middle to high grade) in the LRM, also significantly increased and
decreased the odds ratio of R and S events, respectively. However, the average thickness
of litter layers ordinal variable (TLL) has not exerted significant effect on the odds ratio
of erosion events. Table S-1 and S-2 in supplementary information systematically
describe the whole processes of LRM to evaluate the effect of single factor on odds
ratio of erosion event.
Secondly, we further applied LRM to evaluate the interactive effects of multiple
influencing factors on the odds ratio of R and S events in all restoration vegetation types
(table 7). As to the interactive effect of two soil hydrological properties, the interaction

between low-grade of SHC and increasing-grade of ASM significantly raised the odds

ratio of erosion events. Such that the odds ratio of R and S events affected by the

interactive effects of low-grade of SHC and extreme-grade of ASM were respectively

7.02 and 1.82 times larger than that interactive effects of low-grade of SHC and low-

grade of ASM. Similarly, as to the effect of two vegetation properties, the interactive

effect of low-grade of CRO and increasing-grade of TLL would reduce the odds ratio

of erosion events. Such that the odds ratio of R and S events influenced by the

interaction between low-grade of CRO and high-grade of TLL were respectively only

0.12 and 0.33 times larger than that interactive effects of low-grade of CRO and low-

grade of TLL. Additionally, with respect to the interaction between soil and plant

properties, the interactive effect of low-grade of CRO and increasing-grade of ASM

properties also significantly raised the odds ratio of erosion events. The whole processes

of LRM to evaluate the interactive effect of multiple factors on odds ratio of erosion

event were indicated by the table S-3,4 and 5 in the supplementary information.

Table 5

Table 6

Table 7

Table S-1,2,3,4,5

## 5. Discussion

### 5.1 The integrated probabilistic assessment to erosion stochasticity

The probabilistic attribution and description of stochastic erosion events constituted the framework of integrated probabilistic assessment (IPA).

First, as to one pattern of probabilistic attribution in the IPA, Bayes model supplies a supplementary view and algorithm about how to evaluate the feedback of a result which had stochastically occurred on all possible reasons (Wei and Zhang, 2013). Under the conditions of insufficient information about an occurred result, Bayes model can determine which reasons have the relative greater probability to trigger the occurrence of the result through some prior information. Specific to this study, Bayes model was used to evaluate the probabilistic contribution of four types of I events on one stochastic R ($P(I_k|R)$) and S ($P(I_k|S)$) event generated in each restoration vegetation. Although there were no more specific information about a stochastic soil erosion event, the prior information ($P(R|I_m), P(S|I_m), P(I_m)$) can provide assistance for us to assess the feedback of the stochasticity of soil erosion on different random rainfall events by Bayes model. Meanwhile, ($P(I_k|R)$) and ($P(I_k|S)$) also reflect the different probability threshold values of four rainfall event types triggering soil erosion. Bayes model integrated with total probability theory to systematically quantify the interactive relationship between the stochasticity of precipitation and soil erosion, forming a relative simple and practicable risk assessment of soil erosion event occurring in complex restoration vegetation conditions.

Secondly, as a pattern of probabilistic description in the IPA, Binomial and Poisson

PMFs are two crucial probabilistic functions to characterize many random hydrological
phenomena and to model their ecohydrological effects in natural condition (Eagleson,
1978, Rodriguez-Iturbe et al, 1999, 2001). In this study, the two PMFs were found to
have good simulations of the frequency of times of soil erosion events in three
restoration vegetation types. However, it is necessary and meaningful for the reliability
and accuracy of the IPA to assume whether the two PMFs can both stably and
reasonably simulate the erosion stochasticity at closed-runoff-plot over longer
monitoring period. Therefore, based on above assumption, two important point
estimations methods—the maximum likelihood estimator (MLE) and uniformly
minimum variance unbiased estimator (UMVUE) (Robert et al., 2013)—were applied
to evaluate the stability of erosion stochasticity estimation by means of analyzing the
unbiasedness and consistency of $p_R, p_S, \lambda_R$ and $\lambda_S$. Taking parameter analysis of
random runoff event for example, we defined $X_i$ as the number of times of R event
occurring in some specific restoration vegetation in $i^{\text{th}}$ rainy season ($i = 1,2,3,4$ and $5$).
The five independent and identical (*iid*) random variables satisfies the same and
mutually independent binomial or Poisson PMFs as follows:
$X_1, X_2, \ldots, X_5 \xrightarrow{iid} binomial\ (p_R)$ or $X_1, X_2, \ldots, X_5 \xrightarrow{iid} Poisson\ (\lambda_R)$     (18)
Considering longer monitoring periods, we supposed that the numbers of corresponding
I events (*n*) and rainy seasons (*i*) would approach infinity (*n, i→∞*), and (18) can be
transformed as follow:
$X_1, X_2, \ldots, X_i \xrightarrow{iid} binomial\ (p)$ or $X_1, X_2, \ldots, X_i \xrightarrow{iid} Poisson\ (\lambda)$     (19)
We take MLE and UMVUE methods to search for the best reasonable population
estimators $\hat{p}$ and $\hat{\lambda}$ to approximate the unknown $p$ and $\lambda$ in (19), and finally obtain
more comprehensive stochastic information about the randomness of R event over $i$
rainy seasons. The Appendix B proved that the best estimator $\hat{p}$ in Binomial PMF is
the unbiasedness and consistency of the MLE of $p$. However, proved by the Appendix
C, the best estimator $\hat{\lambda}$ in Poisson PMF have more reliability as it is not only the
unbiasedness and consistency of the MLE of $\lambda$, but also the UMVUE of MLE. The
UMVUE in Poisson PMF implied that lowest variance unbiased estimator can make
the Poisson PMF to be more steadily and accurately stimulate the stochasticity of soil
erosion events over long-term observation than binomial PMF.
Thirdly, besides having better simulation of the stochastic soil erosion events at larger
temporal scale, the Poisson PMF could also be more suitable for simulating the
randomness of S event in the closed-design plot system than that of binomial PMF.
As the hypothesis of Boix-Fayos et al in 2006, the closed runoff-plot design forms
an obstruction to prevent the transportable material from entering the close monitoring
system, which, in particular, lead the transport-limited erosion pattern to gradually
transform into detachment-limited pattern in the closed-plot over time (Boix-Fayos et
al., 2007; Cammerraat, 2002). Consequently, with the extension of monitoring period,
this closed runoff-plot design would cause the sediment more and more difficult to
migrate out of plot, which also reduce the probability of observed S events under the
same precipitation condition. In fact, the effect of closed runoff-plot on stochastic
sediment event could also be successfully implied by the algorithm of Poisson PMF.
Specifically, in order to satisfying the fact that $\lambda=np$ in Poisson PMF is an unknown
constant, the extension of monitoring period could lead to the numbers of times of I
events ($n$) approach infinity, then the probability ($p$) of R or S events generation have
to approach to zero. Above inference coincides with the assumption about the
decreasing of sediment generation in closed-plot system, and further proves that
Poisson PMF could be more reliable to simulate the stochastic erosion events at longer
temporal scale.

**5.2 The effect of influencing factors on erosion stochasticity**
The effects of rainfall, soil and vegetation properties on erosion stochasticity in different
restoration vegetation types were evaluated by LRM. It integrated stochastic rainfall
events with their precipitation and intensity grades, and connected the ecohydrological
functions of soil and plant with their classified hydrological and morphological features.
Just as serving as previous studies (Verheyen and Hermy, 2001a, 2001b; Verheyen et
al., 2003 and Hermy, 2001a; 2001b; Verheyen et al., 2003), LRM in this study explored
the relative importance of morphological features disturbing on the transmission of
stochastic signal of I events into R and S events in different restoration vegetation types.
These disturbances are closed related to the complex hydrological functions owned by
different morphological structures, which finally affect the whole processes of runoff
production and sediment yield (Bautista et al., 2007; Puigdefábregas, 2005).
First, many previous field experiments and mechanism models have proved that
canopy structure has capacity for intercepting intercept precipitation. This specific
hydrological function could potentially prevent the rainfall from directly forming
overland flow or splashing soil surface particles (Liu, 2001; Mohammad and Adam,
2010; Morgan, 2001; Wang et al., 2012). The precipitation retention owned by canopy
structure was regarded as an indispensable positive factor to reduce the soil erosion rate.
Meanwhile, as a crucial complement to understanding hydrological function of canopy
structure, the result of LRM in this study indicated that the higher-grade canopy
structure was a most important morphological feature to reduce the odds ratio of
random soil events in all restoration vegetation types. This result suggests that, the
larger canopy diameter would have relatively stronger capacity for disturbing the
transmission processes of stochastic signal of rainfall on the soil surface than that of
other morphological properties. From the perspective of erosion stochasticity, the
higher-grade canopy structure could finally attribute to the lower probability of R and
S event generation. Therefore, the diversity of canopy structures in different vegetation
types could act a key role on both reducing the intensity and probability of soil erosion
generation.
Secondly, many studies have also discovered that the denser root system distributing
in soil matrix could improve the reinfiltration of the overland (Gyssels et al., 2005).
This reinfiltration process is an effective way to recharge soil water stores when the
overland flow started to occur in hillslopes, which was also an indispensable
contributing factor to reduce the unit area runoff production (Moreno-de las Heras et
al., 2009;Moreno-de las Heras et al., 2010). In this study, the potential reinfilitration
capacity of soil matrix could be positively affected by the saturated hydraulic soil
conductivity (SHC) index. Figure 7 further indicated the distribution patterns of root
system in three restoration vegetation types. Meanwhile, the result of LRM also implied
that the grade of SHC could negatively affect the odds ratio of stochastic erosion event,
which improved the understanding of the hydrological function of root distribution of
plant from the view of erosion randomness. It may suggest that the denser root system
could create more macropores in the subsurface to provide more probability of
reinfiltration of overland flow. This disturbance of overland flow by SHC could reduce
the probability of erosion event generation.

643        Thirdly, the litter layer was proved to act multiple roles on conserving the rainfall,

improving infiltration of throughfall, as well as cushioning the splashing of raindrop
(Gyssels et al., 2005; Munoz-Robles et al., 2011;Geißler et al., 2012). Therefore, the
thicker litter layer in T2 (figure 7) probably has stronger capacity for conserving and
infiltrating throughfall, as well as inhibiting splash erosion than that of other restoration
vegetation types (Woods and Balfour, 2010). Although the result of LRM indicated that
there was no significant correlation between the litter layer thickness (TLL) and the
odds ratio of soil erosion (table 6), the interactive effect of TLL and CRO significantly
affect the odds ratio of stochastic erosion events (table 7). The interaction result implied
that, under the relative low-grade CRO condition, the higher-grade TLL could have
stronger disturbance on the transmission of stochastic signals of rainfall to improve the
throughfall absorption to reduce the probability of splash or sheet erosion occurrence.

655        Additionally, table 7 explored more interactive effects of the soil and plant properties

on odds ratio of random runoff and sediment event. These explorations suggested that
the interactions between soil and vegetation properties formed more complex
hydrological functions to affect the stochastic soil erosion event during whole
ecohydrological processes in semi-arid environment (Ludwig et al., 2005).
Although the hydrological-trait of vegetation acted as core roles on reducing the soil
erosion depending on the mechanical properties of their morphological structures (Zhu
et al., 2015), the LRM analysis in this study further illuminated that these hydrological-
trait morphological structure of vegetation may also play an important role on affecting
the stochasticity of soil erosion. Actually, the different stochasticity of soil erosion in
three restoration vegetation types reflected the different extents of disturbance of
vegetation types on the transmission of stochastic signals of rainfall into soil-plant
systems. Therefore, the relative smaller canopy structure, thinner litter layer, and
shallower root system in T3 have relatively weaker capacity to disturbing the stochastic
signal of rainfall than that of T1 and T2 with obvious hydrological-trait morphological
structures (figure 7). The effect of diverse morphological structures on stochasticity of
soil erosion was a meaningful complement to studying on the hydrological functions of
restoration vegetation types in semi-arid environment.

674                Figure 7

675                Table 6

676                Table 7




**5.3 The implication of integrated probabilistic assessment**

The integrated probabilistic assessment (IPA) could be an important complement to expand on the understanding of hydrological function existing in vegetation types. The hydrological-trait of morphological structures owned by different plants is closely related to the function of vegetation-driven in affecting the intensity of erosion events. The vegetation-driven-spatial-heterogeneity (VDSH) theory (Puigdefábregas, 2005) could be regarded as a clear concise summary to emphasize the dominant role of vegetation in restructuring soil erosion processes. It reflected the effect of spatial distribution patterns of vegetation on their corresponding hydrological functions on controlling erosion rate in patch, stand, and even regional scales. Therefore, VDSH theory has provided an innovative view to investigating the soil erosion and other ecohydrological phenomena affected by vegetation (Sanchez and Puigdefábregas, 1994;Puigdefábregas, 1998;Boer and Puigdefábregas, 2005). In the study, depending on the long-term experimental data and fundamental probability theories, the IPA concentrated on the hydrological function of vegetation-driven in affecting the randomness of erosion events rather than the erosion rate. It could enrich the comprehension of hydrological function of vegetation morphological structure on soil erosion phenomena, and also be effective complement for application of VDSH theory on interpreting the stochastic erosion events.

Additionally, in our study, the IPA could also provide a new framework for practitioners to develop restoration strategies which focused on controlling the risk of erosion generation rather than only on reducing erosion rate. The framework contains

three stages including construction of stochastic environment, description of random
erosion events, and evaluation of probabilistic attribution (figure 8).
The first stage in the framework aims to build a unified platform to describe the
stochasticity of different hydrological phenomena closely related to the erosion event.
This stage generally investigates the stochastic background under which soil erosion
generation, which is also an indispensable precondition for quantifying the probability
of R and S in stage II. The second stage is designed to construct a phased adjustment of
monitoring processes based on the principle of Bayes theory as well as on the parameter
analysis of Binomial and Poisson models. In this phased-adjustment monitoring, the
Bayes, Binomial and Poisson models were applied on simulating the randomness of
erosion events in short-term, mid-term and long-term monitoring stages, respectively.
This model-driven monitoring approach could be regarded as a more reasonable method
to explore the complexity of stochastic erosion events in larger temporal scales, but also
provide a new perspective for researchers to more effectively evaluate the stochasticity
of erosion events in stage III. The objective of stage III is to assess the probabilistic
attribution of rainfall, soil and vegetation properties on erosion events generation. This
probabilistic attribution evaluation by LRM, could develop the restoration strategies for
more effectively selecting vegetation types with stronger capacity for reducing the
erosion risk, and finally improve the management of soil and water conservation in a
semi-arid environment.
As a result, this stochasticity-based restoration strategy was developed by a
combination of experimental data with multiple probabilistic theories to deal with the
soil erosion randomness under complex stochastic environment. It is different from the
trait-based restoration scheme derived from the functional diversity of vegetation
community to reduce the soil erosion rate (Zhu et al., 2015; Baetas et al., 2009).
Meanwhile, with the increase of monitoring duration, more stochastic information of
erosion events could be added into the IPA framework. This addition could finally fulfil
the self-renewal and self-adjustment of the IPA to improve the restoration strategy for
selecting more reasonable vegetation types with stronger capacity for controlling
erosion risk in long term. Therefore, the IPA framework containing three stages could
translate the event-driven erosion stochasticity into restoration strategies concentrating
on erosion randomness, which may be a meaningful complement for restoration
management in a semi-arid environment.

736                           Figure 8


**6. Conclusion**
In this study, we applied an integrated probabilistic assessment (IPA) to describe,
simulate and evaluate the stochasticity of soil erosion in three restoration vegetation
types in the Loess Plateau of China, and draw the following conclusions:
(1) In the IPA, the OCIRS was an innovative event-driven system to standardize the
definition of hydrological random events, which is also a foundation for quantifying
the stochasticity of soil erosion events under complex environment conditions.
(2) Both of binomial and Poisson PMFs in the IPA could simulate the probability
distribution of the numbers of runoff and sediment events in all restoration
vegetation types. However, Poisson PFM could more effectively simulate the
stochasticity of soil erosion at larger temporal scales.
(3) The difference of morphological structures in restoration vegetation types is the
main source of different stochasticity of soil erosion from T1 to T3 under same
rainfall condition. Larger canopy, thicker litter layer and denser root distribution
could more effectively affect the transmission of stochastic signal of rainfall into
soil erosion.
The IPA is an important complement to developing restoration strategies to improve
the understanding of stochasticity of erosion generation rather than only of the intensity
of erosion event. It could also be meaningful to researchers and practitioners to evaluate
the efficacy of soil control practices in a semi-arid environment.

**Appendix A.   The transformation from binominal to Poisson PMF**
Let $p = \frac{\lambda}{n}$, then:
$pmf_{Xbin}(x) = \binom{n}{x}p^x(1-p)^{n-x} = \frac{n!}{x!(n-x)!} \cdot \left(\frac{\lambda}{n}\right)^x \cdot \left(1 - \frac{\lambda}{n}\right)^{n-x}$
$= \frac{\lambda!}{x!} \cdot \frac{n(n-1)(n-2)\cdots 1}{(n-x)(n-x-1)\cdots 1} \cdot \frac{1}{n^x} \cdot \left(1 - \frac{\lambda}{n}\right)^{n-x}$
$= \frac{\lambda!}{x!} \cdot 1 \cdot \left(1 - \frac{1}{n}\right) \cdot \left(1 - \frac{2}{n}\right) \cdots \left(1 - \frac{x-1}{n}\right) \cdot \left(1 + \frac{-\lambda}{n}\right)^n \cdot \left(1 - \frac{\lambda}{n}\right)^{-x}$   (A1)
In equation (A1), when $n \to \infty$, and $x, \lambda$ is finite and constant, then
$\lim_{n\to\infty}(1 - \frac{1}{n}) = \cdots = \lim_{n\to\infty}\left(1 - \frac{x-1}{n}\right) = \lim_{n\to\infty}\left(1 - \frac{\lambda}{n}\right)^{-x} = 1$   (A2)
And
$\lim_{n\to\infty}\left(1 + \frac{-\lambda}{n}\right)^n = e^{-\lambda}$   (A3)
And according to equation (A2) and (A3), the equation (A1) can be transformed as:

$$\lim_{n \to \infty}\left[\frac{n!}{x!\,(n-x)!} \cdot \left(\frac{\lambda}{n}\right)^x \cdot \left(1 - \frac{\lambda}{n}\right)^{n-x}\right] = \frac{\lambda^x e^{-\lambda}}{x!} \quad x = 0,1,2,\dots \tag{A4}$$


or

$$pmf_{Xbin}(x) \xrightarrow{n \to \infty} \frac{\lambda^x e^{-\lambda}}{x!} = pmf_{Xpoi}(x) \tag{A5}$$



## 773 Appendix B. Parameter estimation of $p$ in Poisson PMF

### 774 (1) Derivatization of the MLE $\widehat{p}$

Let the random sample $X_1, X_2, \dots, X_i \xrightarrow{iid} pmf_{Xbin}(p)$ and assume the binomial
distribution as:

$$P(X = x_i) = \binom{m}{x_i} p^{x_i}(1-p)^{m-x_i} \tag{B1}$$


The likelihood function $L(p)$ is joint binomial PDF with parameter $p$ as follow:

$$L(p) = f_X(X_1, \dots, X_n, p) = \prod_{i=1}^{n}\binom{m}{x_i} p^{\sum_{i=1}^{n} X_i}(1-p)^{(mn - \sum_{i=1}^{n} X_i)} \tag{B2}$$


By taking logs on both side of equation (B2):

$$lnL(p) = ln\left(\prod_{i=1}^{n}\binom{m}{x_i}\right) + \sum_{i=1}^{n} X_i\, ln\,p + \left(mn - \sum_{i=1}^{n} X_i\right)ln(1-p) \tag{B3}$$


And differentiating with respect to $p$ in $lnL(P)$ and let the result be zero:

$$\frac{\partial lnL(p)}{\partial p} = \frac{\sum_{i=1}^{n} X_i}{p} - \frac{(mn - \sum_{i=1}^{n} X_i)}{(1-p)} = 0 \tag{B4}$$


Solution $\hat{p} = \frac{\sum_{i=1}^{n} X_i}{mn}$, let $m = n, \implies \hat{p} = \frac{\overline{X}}{n}$
Therefore, $\hat{p} = \frac{\overline{X}}{n}$ is the MLE of population parameter $p$ in binomial PMF model.

### 787 (2) Discussion of the unbiasedness and consistency of $\widehat{p}$

Let $E_p(\hat{p})$ be the expectation of M.L.E $\hat{p}$ when population parameter $p$ is true in
random sample which is $X_1, X_2, \ldots, X_i \xrightarrow{iid} pmf_{Xbin}(p)$, then
$$E_p(\hat{p}) = E_P(\overline{X}/n) = \frac{1}{n^2} \sum_{i=1}^{n} E_P(X_i) = \frac{1}{n^2} n^2 p = p \qquad \text{(B5)}$$
Which proved that MLE $\hat{p} = \frac{\overline{X}}{n}$ is a unbiased estimator for $p$. And furthermore then
let $Var_p(\hat{p})$ be the variance of $\hat{p}$ when population $p$ is true.
$$Var_p(\hat{p}) = Var_p\left(\sum_{i=1}^{n} X_i/n^2\right) = \frac{1}{n^4} \sum_{i=1}^{n} Var_p(X_i) = \frac{p(1-p)}{n^2} \qquad \text{(B6)}$$
As the $n$ approaches to infinite:
$$\lim_{n \to \infty} Var_p(\hat{p}) = \lim_{n \to \infty} \left(\frac{p(1-p)}{n^2}\right) = 0 \qquad \text{(B7)}$$
Equation (B5)~(B7) satisfied the theme of weak law of larger number, which lead the
$\hat{p} = \frac{\overline{X}}{n}$ is probabilistic converge to population parameter $p$:
$$\lim_{n \to \infty} P(|\hat{p} - p| \geq \varepsilon) = 0, \text{for all } \varepsilon > 0 \qquad \text{(B8)}$$
Consequently, the unbiased MLE $\hat{p} = \frac{\overline{X}}{n}$ is consistent for $p$.

**Appendix C.    Parameter estimation of $\lambda$ in Poisson PMF**
**(1)   Derivatization of the MLE $\hat{\lambda}$**
Let the random sample $X_1, X_2, \ldots, X_i \xrightarrow{iid} pmf_{Xpoi}(\lambda)$, and assume the poisson
distribution as:
$$pmf_{Xpoi}(x_i) = \frac{\lambda^{x_i} e^{-\lambda}}{x_i!} \qquad \text{(C1)}$$
The likelihood function $L(\lambda)$ is joint PDF with parameter $\lambda$ as follow:
$$L(\lambda) = f_X(X_1, \ldots, X_n, \lambda) = f(X_1, \lambda) \times \cdots \times f(X_n, \lambda) = \prod_{i=1}^{n} \frac{\lambda^{x_i} e^{-\lambda}}{x_i!} \qquad \text{(C2)}$$
Taking logs on $L(\lambda)$ in equation (B4) and differentiating logarithm function with
respect to $\lambda$:
$$\frac{\partial lnL(\lambda)}{\partial \lambda} = \frac{\partial (\prod_{i=1}^{n} \frac{\lambda^{x_i} e^{-\lambda}}{x_i!})}{\partial \lambda} = -n \frac{\lambda^{\sum_{i=1}^{n} X_i}}{(x_1 x_2 \cdots x_n)!} e^{-n\lambda} + \frac{\sum_{i=1}^{n} X_i \lambda^{(-1+\sum_{i=1}^{n} X_i)}}{(x_1 x_2 \cdots x_n)!} \quad (C3)$$
Let the equation (C3) equal to zero, and has solution:
$$\hat{\lambda} = \frac{1}{n} \sum_{i=1}^{n} X_i = \overline{X} \quad (C4)$$
Therefore, $\hat{\lambda} = \overline{X}$ is the MLE of population parameter $\lambda$ in Poisson PMF model.

**(2) Discussion of the unbiasedness and consistency of $\hat{\lambda}$**
Let $E_\lambda(\hat{\lambda})$ be the expectation of MLE $\hat{\lambda}$ when population parameter $\lambda$ is true in
random sample $X_1, X_2, \ldots, X_i \xrightarrow{iid} pmf_{Xpoi}(\lambda)$, then:
$$E_\lambda(\hat{\lambda}) = E_\lambda(\overline{X}) = \frac{1}{n^2} \sum_{i=1}^{n} E_\lambda(X_i) = \frac{1}{n} n\lambda = \lambda \quad (C5)$$
which proved that MLE $\hat{\lambda} = \overline{X}$ is a unbiased estimator for $\lambda$. Meanwhile, let $Var_\lambda(\hat{\lambda})$
be the variance of MLE $\hat{\lambda}$ when population parameter $\lambda$ is true
$$Var_\lambda(\hat{\lambda}) = Var_\lambda(\overline{X}) = Var_\lambda\left(\sum_{i=1}^{n} X_i / n^2\right) = \frac{1}{n^4} \sum_{i=1}^{n} Var_\lambda(X_i) = \frac{\lambda}{n} \quad (C6)$$
And
$$\lim_{n \to \infty} Var_\lambda(\hat{\lambda}) = \lim_{n \to \infty} \left(\frac{\lambda}{n}\right) = 0 \quad (C7)$$
According to the weak law of large number theme, equation (B7, B8, C1) lead that
unbiased MLE $\hat{\lambda} = \overline{X}$ is probabilistic converge to $\lambda$:
$$\lim_{n \to \infty} P(|\hat{\lambda} - \lambda| \geq \varepsilon) = 0, \text{for all } \varepsilon > 0 \quad (C8)$$
Therefore, MLE $\hat{\lambda} = \overline{X}$ is consistent for population parameter $\lambda$.

**(3) Determination of UMVUE $\hat{\lambda}$ of population parameter**
Firstly, MLE $\hat{\lambda} = \overline{X}$ is an unbiased estimator of parameter $\lambda$ which is the
precondition of UMVUE determination. Secondly, by using Cramer-Rao lower bound

to check whether the unbiased MLE was UMVUE or not. Then we have:

$$lnf_X(X, \lambda) = -lnx! + xln\,\lambda - \lambda \tag{C9}$$

$$\frac{\partial(lnf_X(X, \lambda))}{\partial\lambda} = \frac{x}{\lambda} - 1 \tag{C10}$$

And

$$\frac{\partial^2 lnf_X(X, \lambda)}{\partial\lambda^2} = \frac{\partial(\frac{x}{\lambda} - 1)}{\lambda} = -\frac{x}{\lambda^2} \tag{C11}$$

Accordingly the expectation of equation (C11) when the population parameter $\lambda$ is true:

$$E_\lambda\left[\frac{\partial^2 lnf_X(X, \lambda)}{\partial\lambda^2}\right] = E_\lambda\left(-\frac{X}{\lambda^2}\right) = -\frac{1}{\lambda^2}E_\lambda(X) = -\frac{\lambda}{\lambda^2} = -\frac{1}{\lambda} \tag{C12}$$

So the Cramer-Rao lower bound (CRLB) is

$$CRLB = \frac{1}{-nE_\lambda\left[\frac{\partial^2 lnf_X(X, \lambda)}{\partial\lambda^2}\right]} = \frac{1}{-n\cdot(-\frac{1}{\lambda})} = \frac{\lambda}{n} = Var_\lambda(\hat{\lambda}) = Var_\lambda(\overline{X}) \tag{C13}$$

Consequently, MLE $\hat{\lambda} = \overline{X}$ is UMVUE of population parameter $\lambda$.

**Acknowledgement**

This work was funded by the National Natural Science Foundation of China (No. 41390464) and the National Key Research and Development Program (No. 2016YFC0501602). We specially thank for associated editor, and two reviewers whose suggestions and advices improve the quality of this study, we also thank professor Chen Lin-An with National Chiao Tung University (NCTU) for his great help on the mathematical statistical inference in this manuscript, and thank Liu Yu, Liu Jianbo and Wang Jian for their support for soil erosion monitoring.

**Figure captions**

Figure 1    The construction of OCIRS system : (a) a flow chart to determine all random event types
in OCIRS framework; (b) the different combining patterns of rainfall and non-rainfall events in three
consecutive days to form ten observed random event sequences on five rainy seasons; (c) Venn
diagram to reveal the relationship among all random events types in OCIRS framework.

Figure 2    Study area and experimental design: (a) location of the Yangjuangou Catchment; (b)
three restoration vegetation types including *Armeniaca sibirica* (T1), *Spiraea pubescens* (T2), and
*Artemisia copria* (T3); (c) the dynamic measurement of soil moisture and data collection to provide
the information about average antecedent soil moisture; (d) the measurement of field saturated
hydraulic conductivity to determine the average infiltration capability; (e): the investigation of
morphological properties of restoration vegetation by setting quadrats

Figure 3    The probability distribution of different random rainfall event types (Iw, Is, Il, and Ie)
and random non-rainfall event types (Ch and Cd) at monthly and seasonal scales from rainy season
of 2008 to 2012.

Figure 4    The probability distribution of random runoff and sediment events generating in three
restoration vegetation types at monthly and seasonal scales from rainy season of 2008 to 2012, the
Arabic numbers and letter "T" on the abscissa indicate the month and season respectively, the same
as follow figures

Figure 5    The comparison between simulation of stochasticity of runoff and sediment events by
Binomial and Poisson PMFs and the observed frequencies of numbers of times of soil erosion events
in three restoration vegetation type, Exp_B and Exp_P indicates the simulated values in Binomial
and Poisson PMF respectively, and the histogram represents the observed values.

Figure 6    The distribution of probabilistic contribution of four random rainfall event types on
anyone runoff or sediment event stochastically generating in three restoration vegetation types at
monthly and seasonal scales from rainy season of 2008 to 2012

Figure 7    Morphological properties of three restoration vegetation types including the thickness
of litter layer, the distribution of root system. The dashed lines indicates the diameter and depth of
soil samples with approximating 10 cm and 30 cm respectively.

Figure 8    The framework of integrated probabilistic assessment for soil erosion monitoring and
restoration strategies

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

Tables

Table 1    The summary of main researches on the stochasticity of soil erosion rate and the stochasticity of factors to affect the soil erosion rate

| [a]Stochasticity (Uncertainty) | [b]Approach or method | [c]Driven types | Main Hydrological behaviors | Main Influencing factors | Spatiotemporal Scale | Reference |
|---|---|---|---|---|---|---|
| Stochasticity of soil erosion rate | | | | | | |
| Runoff connectivity | Probabilistic model Conceptual model | (1)Data-Mapping (2)Theory | Infiltration processes Precipitation | Topography Soil depth | Hillslope scale in USA | Janzen, D., and McDonnell, J 2015 |
| Runoff processes | Probabilistic model Conceptual model | (1)Simulation (2)Theory | Infiltration processes Precipitation | Topography | | Janzen, D., and McDonnell, J 2015 |
| Runoff production | Probabilistic model Conceptual model | (1)Theory (2)Simulation | Runoff absorption Water storage Infiltration capacity | Soil moisture Evaporation Recharge | Point and basin scale | Moore, 2007 |
| Flood prediction and runoff | Probabilistic model Multivariate analysis | (1)Simulation (2)Data-Calibration | Stochastic rainfall process | Parameters in rainfall-runoff model | Multiple catchment scales in Iran | Yazdi, J. et al., 2014 |
| Rainfall and runoff processes | Probabilistic model hydrological mechanism | (1)Simulation (2)Random event (3)Theory | Soil storage | Given climate regime hydraulic conductivity landform development | Hillslope scale | Freeze, 1980 |
| Erosion rate | Probabilistic model Mechanical mechanism | (1)Data-Calculation (2)stochastic forcing | | Bed shear stress Critical shear stress | Laboratory scales in Netherlands | Prooijen and Winterwerp, 2010 |
| Erosion rate | Physical model Probabilistic model Conceptual model | (1)Theory (2)Simulation | Simulated near-bed flow | Soil structure Oscillating flow | | Sidorchuk, 2005 |

| | | | | | | |
|---|---|---|---|---|---|---|
| Erosion risk | Empirical model Geo-statistics | (1)Data-Mapping | Erosive precipitation | Factors in RUSLE | Annual and Reginal scales in China | Jiang et al., 2012 |
| Uncertainty of soil loss | Empirical model Geo-statistics Error analysis | (1)Simulation (2)Data-calibration | Erosive precipitation Runoff and sediment | Spatiotemporal Rainfall erosivity distribution | Annual time and catchment scale in USA | Wang et al., 2002 |
| Uncertainty and variability of erosion rate | Empirical model | (1)Hypotheses (2)Data-calculation | Total rainfall volume and 30-minute rainfall intensity | Stochastic environment conditions Scale effect | | Kim et al., 2016 |
| Stochasticity of factors to affect soil erosion rate | | | | | | |
| Soil moisture related to soil erosion | Probabilistic model Physical model | (1)Hypotheses, (2)Simulation (3)Theory | Precipitation Evapotranspiration | Temporal patterns of rainfall property | Daily time and Hillslope scale in | Ridolfi et al., 2003 |
| Antecedent soil moisture related to soil erosion | Probabilistic model Physical model | (1)Data-Mapping (2)Theory | Runoff response Infiltration processes | | Daily time and multiple catchment scales in Spain | Castillo et al., 2003 |
| Stochastic rainfall related to flood and runoff | Probabilistic model Conceptual model | (1)Data-Calibration (2)Random event (3)Hypothesis | Stochastic storm Runoff and flood | Parameters in Peak flow models | Hourly-daily time and multiple catchment scales in Germany | Haberlandt and Radtke, 2014 |
| Stochastic rainfall related to runoff and erosion | Physical model Empirical model | (1)Simulation (2)Data-calibration | Overland/channel flow Erosion transport Precipitation | Spatiotemporal rainfall distribution | Seasonal and annual time catchment scale in USA | Lopes, 1996 |
| Uncertainty of soil erodibility | Empirical model Geo-statistics | (1)Simulation (2)Data-Mapping | | Spatiotemporal soil types, depth and parent material | Regional scales in USA | Wang et al., 2001 |
| Stochastic rainfall | Probabilistic model | (1)Data-calibration | Sewer overflows | Rainfall depth and | Seasonal and annual | Andres- |

| related to runoff | Conceptual model Physical model | (2)Theory | | duration, conditions | climate time scales in Spain | catchment | Domenech et al., 2010 |
|---|---|---|---|---|---|---|---|

a: the main contents of different studies focusing on the stochasticity (uncertainty) of soil erosion and its influencing factors

b: the main statistical methods or different types of mathematic and physical models to be employed to describe and analyze the stochasticity of soil erosion

c: the main properties of analyzing framework in the different studies and the characteristics of data application on the evaluation of stochasticity of soil erosion





Table 2   Definition and explanation of all random events in OCIRS

| symbol | Physical meaning of random event types | Probabilistic meaning of random event types | Influencing factors and implication |
|---|---|---|---|
| O | observation events with time step ranging from 0 to 72 hours, including non-rainfall and rainfall events | random events composing the sample space of OCIRS system. The probability $P(O) = 1$ | indicating the general stochastic weather conditions over rainy seasons |
| C | non-rainfall events with time step ranging from 0 to 24 hours, including sunny or cloudy weather condition at hour or day scales | random events, the probability of C events is the ratio of numbers of C events to O events $C \subset O, 0 \leq P(C) \leq P(O) = 1$ | implying the extent of evaporation or potential evapotranspiration in weather condition. |
| Cd | non-rainfall events with time step being 24 hours, including observed sunny or cloudy at day scale | random events composing the subset of C events, $Cd \subseteq C, 0 \leq P(Cd) \leq P(C)$ | implying the duration of evaporation or evapotranspiration at day scale |
| Ch | non-rainfall events with time step being less than 24hours, including observed sunny or cloudy at hour scales which intercepted by rainfall events within a day | random events composing the subset of C events, the intersection of Ch and Cd is null, $Ch \subseteq C, Cd \cup Ch = C, Cd \cap Ch = \emptyset, 0 \leq P(Ch) \leq P(C)$ | influenced by the frequency of rainfall events generation, and implying the alternation of sunny and rainy in a day |
| I | an individual rainfall event with different precipitation, intensity and duration ranging from 0 to 72 hours, the time interval between two I events is more than 6 hours | random events, the probability of I event is ratio of numbers of I events to O events over observation $I \subset O, I \cup C = O, I \cap C = \emptyset, 0 \leq P(I) \leq P(O) = 1$ | a driven force of soil erosion, which could be intercepted by vegetation and transformed into throughfall |

| | | | |
|---|---|---|---|
| Ie | an extreme longest individual rainfall event whose average precipitation, intensity and duration were 96.6 mm, 0.022 mm/min, and 73 hours, respectively. | random events composing the subset of I events, Ie ⊆ I, 0 ≤ P(Ie) ≤ P(I) | rainfall events with low intensity and longest duration, inclining to infiltration-excess runoff generation |
| Il | a second longest individual rainfall events types whose average precipitation, intensity and duration were 47.3 mm, 0.027 mm/min, and 30 hours, respectively. | random events composing the subset of I events, the intersection of Il and Ie is null, Il ⊆ I, Il ∩ Ie = ∅, 0 ≤ P(Il) ≤ P(I) | rainfall events with low intensity and long duration, inclining to infiltration-excess runoff generation |
| Is | A rainfall event type spanning two days whose average precipitation, intensity and duration were 22.7 mm, 0.042 mm/min, and 10 hours, respectively | random events composing the subset of I events, Is ⊆ I, Is ∩ Il ∩ Ie = ∅, 0 ≤ P(Il) ≤ P(I) | rainfall events with strongest rainfall intensity in middle duration, inclining to runoff and sediment generation |
| Iw | a rainfall event type generating within a day whose average precipitation, intensity and duration were 9.8 mm, 0.045 mm/min, and 5 hours, respectively. it usually generates several times within one day. | random events composing the subset of I events, Iw ⊆ I, Iw ∩ Is ∩ Il ∩ Ie = ∅, Iw ∪ Is ∪ Il ∪ Ie = I, 0 ≤ P(Iw) ≤ P(I) | rainfall events with fewest and shortest precipitation and duration, which is different to trigger soil erosion |
| R | runoff event type generating on vegetation land types, it occurs on rainfall processes, and its duration is negligible | random events responding to I events, R ⊂ I, R ∩ C = ∅, 0 ≤ P(R) < P(I) | influenced by rainfall and vegetation properties. |
| S | sediment event occurring on vegetation land types, it occurs on runoff processes, and its duration is negligible | random events responding to R events, S ⊂ R ⊂ I, S ∩ C = ∅, 0 ≤ P(S) ≤ P(R) < P(I) | driven by R events, and affected by rainfall and vegetation properties. |








Table 3 Main characteristics of four types of random rainfall event over five rainy seasons

| Rainy season | Rainfall event types | Average precipitation (mm) | Average intensity (mm/min) | Average duration (hour) |
|---|---|---|---|---|
| 2008 | Iw | 16.7 | 0.122 | 2.3 |
| | Is | 19.2 | 0.066 | 4.8 |
| | Il | 53.2 | 0.032 | 27.7 |
| | Ie | 96.6 | 0.022 | 73.2 |
| 2009 | Iw | 9.0 | 0.027 | 5.6 |
| | Is | 35.4 | 0.059 | 10.0 |
| | Il | 47.9 | 0.032 | 24.9 |
| | Ie | × | × | × |
| 2010 | Iw | 9.0 | 0.018 | 8.3 |
| | Is | 7.6 | 0.012 | 10.6 |
| | Il | × | × | × |
| | Ie | × | × | × |
| 2011 | Iw | 3.3 | 0.031 | 1.8 |
| | Is | 21.5 | 0.040 | 9.0 |
| | Il | 42.5 | 0.020 | 35.4 |
| | Ie | × | × | × |
| 2012 | Iw | 10.8 | 0.028 | 6.4 |
| | Is | 30.0 | 0.031 | 16.1 |
| | Il | 45.5 | 0.023 | 33.0 |
| | Ie | × | × | × |
| Average | Iw | 9.8 | 0.045 | 4.9 |
| | Is | 22.7 | 0.042 | 10.1 |
| | Il | 47.3 | 0.027 | 30.3 |
| | Ie | 96.6 | 0.022 | 73.2 |


Table 4    Basic properties of soil, vegetation and erosion in different restoration vegetation types

| Basic properties of different vegetation types | [h]N | Restoration vegetation types | | |
|---|---|---|---|---|
| | | *Armeniaca sibirica* Type 1 (T1) | *Spiraea pubescens* Type 2 (T2) | *Artemisia copria* Type3 (T3) |
| Topography property | | | | |
| Slope aspect | 9 | Southwest | Southwest | Southwest |
| Slope gradation (%) | 9 | ≈26.8 | ≈26.8 | ≈26.8 |
| Slope size for each (m) | 9 | 3×10 | 3×10 | 3×10 |
| Soil property | | | | |
| [a]DBD (g cm$^{-3}$) | 30 | 1.28±0.08 | 1.16±0.12 | 1.23±0.10 |
| Clay (%) | 30 | 11.07±2.43 | 11.98±3.05 | 9.54±1.48 |
| Silt (%) | 30 | 26.11±1.50 | 25.24±3.84 | 26.72±2.87 |
| Sand (%) | 30 | 62.82±0.94 | 62.78±4.51 | 63.74±3.24 |
| [b]Texture type | | Sandy loam | Sandy loam | Sandy loam |
| [c]SHC (cm min$^{-1}$) | 20 | 0.46±0.82(a) | 2.22±0.66(b) | 0.50±0.60(a) |
| [d]SOM (%) | 30 | 1.28±0.63(a) | 0.98±0.15(b) | 0.90±0.09(b) |
| Vegetation property | | | | |
| Restoration years | 9 | 20 | 20 | 20 |
| Crown diameters (cm) | 27 | 211.6±15.4(c) | 80.5±4.5(b) | 64.1±6.3(a) |
| Litter layer (cm) | 30 | 1.2±0.3(a) | 3.4±1.8(b) | 1.8±0.5(a) |
| Height (cm) | 27 | 256.3±11.1(c) | 128.3±8.3(b) | 61.8±1.1(a) |
| LAI | 27 | × | 2.31 | 1.78 |
| [e]Ave. Coverage (%) | 27 | 85 | 90 | 90 |
| Rainfall/Erosion property | | | | |
| Times of rainfall events | | | 130 | |
| Times of runoff events | | 30/30/30 | 45/45/45 | 45/45/45 |
| Times of sediment events | | 13/13/13 | 19/19/19 | 32/32/32 |
| [f]Ave. runoff depth (cm) | | 0.012(a) | 0.014(a) | 0.083(b) |
| [g]Ave. sediment amount (g) | | 5.8(a) | 6.8(a) | 25.7(b) |

a: dry bulk density; b: texture type is determined by textural triangle method based on USDA; c: field saturated hydraulic conductivity, and all the values with same letter in each row indicates non-significant difference at α=0.05 which is the same as follow rows; d: soil organic matter; e: average coverage of three restoration vegetation types over five rainy seasons; f: average runoff depth in restoration types over rainy seasons; g: average sediment yield in restoration types over rainy seasons; h: sample number.








Table 5     The definition and classification of properties of rainfall soil and plant ordinal variables

| Ordinal variable | Physical meaning of classified influencing factors | Standard of influencing factor classification | | | |
|---|---|---|---|---|---|
| | | Low (L) | Middle (M) | High (H) | Extreme (E) |
| PREC | classified precipitation variable of a single random rainfall event | 0~15 mm | 15~30 mm | 30~60 mm | >60 mm |
| INT | classified intensity variable of a single random rainfall event | 0~0.025 mm/min | 0.025~0.05 mm/min | 0.05~0.1 mm/min | >0.1 mm/min |
| ASM | classified variable of the antecedent soil moisture | 0~5 % | 5~10 % | 10~20 % | >20 % |
| SHC | classified variable of the filed saturated hydraulic conductivity | 0~1 cm/min | × | >1 cm/min | × |
| CRO | classified variable of the average crown width in vegetation types | 0~60 cm | 60~80 cm | >80 cm | × |
| TLL | classified variable of the average thickness of litter layers | 0～2 cm | × | >2 cm | × |
| $Y_R$ | dichotomous dependent variable to indicate whether a random runoff event has generation or not | If $Y_R$ =1, it means that a random runoff event has generated; If $Y_R$ =0, it means that a random runoff event has not generated | | | |
| $Y_S$ | dichotomous dependent variable to indicate whether a random sediment event has generation or not | If $Y_S$ =1, it means that a random sediment event has generated; If $Y_S$ =0, it means that a random sediment event has not generated | | | |


Table 6    Logistic regression model to analysis the single effect of rainfall, plant and soil ordinal variable on the erosion events presence/absence in all restoration vegetation types

| Grade levels | PREC (Low) | INT (Low) | ASM (Low) | SHC (Low) | CRO (Low) | TLL (Low) |
|---|---|---|---|---|---|---|
| Odds ratio of all random runoff events | | | | | | |
| Extreme | [a] $\times^{NS}$ | [b] 90.91*** | [c] 2.19* | Null | Null | Null |
| High | $\times^{NS}$ | 32.26*** | 2.01* | [d] 0.85* | [e] $7.53 \times 10^{-3}$** | [f] $\times^{NS}$ |
| Middle | $\times^{NS}$ | 2.09* | 1.59* | Null | $7.17 \times 10^{-2}$** | Null |
| Odds ratio of all random sediment events | | | | | | |
| Extreme | 142.85*** | 166.67*** | 15.40* | Null | Null | Null |
| High | 16.95** | 125.00*** | 13.79** | 0.78* | $6.27 \times 10^{-3}$** | $\times^{NS}$ |
| Middle | 6.09** | 34.48*** | 6.36* | Null | $2.55 \times 10^{-2}$** | Null |

a: making the low-grade of PREC ordinal variable as reference, the odds ratio of all random runoff event in extreme-grade of PREC is not significantly larger than that of low-grade of PREC; b: making the low-grade of INT ordinal variable as reference, the odds ratio of all random runoff events in extreme-grade of INT is 90.91 times significantly larger than that of low-grade of INT, under the controlled PREC condition with $P \leq 0.001$; c: making the low-grade of ASM ordinal variable as reference, the odds ratio of all random runoff events in extreme-grade of ASM is 2.19 times significantly larger than that of low-grade of ASM, under the controlled PREC and INT condition with $P \leq 0.1$; d: making the low-grade of SHC ordinal variable as reference, the odds ratio of all random runoff events in high-grade of SHC is 0.85 times significantly larger than that of low-grade of SHC, under the controlled PREC, INT and ASM condition with $P \leq 0.1$; e: making the low-grade of CRO ordinal variable as reference, the odds ratio of all random runoff events in high-grade of CRO is $7.53 \times 10^{-3}$ larger than that of low-grade of CRO, under the controlled PREC, INT, ASM and SHC condition with $P \leq 0.01$; f: making the low-grade of TLL ordinal variable as reference, the odds ratio of all random runoff event in high-grade of TLL is not significantly larger than that of low-grade of TLL, under the controlled PREC, INT, ASM, SHC and CRO condition. (Wald test statistic is applied to test the significant of odds ratio *** $P \leq 0.001$, ** $P \leq 0.01$, * $P \leq 0.1$, NS: not significant, $\times^{NS}$: the nonsignificant value cannot be estimated)

Table 7    Logistic regression model to analysis the interactive effect of rainfall, plant and soil
ordinal variables on the erosion events presence/absence in all restoration vegetation types

| Grade levels | Reference of grade levels | Soil_ASM | | | | Plant_TLL | |
|---|---|---|---|---|---|---|---|
| | | ASM (low) | ASM (middle) | ASM (high) | ASM (extreme) | TLL (low) | TLL (high) |
| | | Odds ratio of all random runoff events | | | | | |
| Soil_SHC | SHC (low) | Ref. | [a]2.23$^{NS}$ | 3.19$^{NS}$ | 7.02* | Null | Null |
| Plant_TLL | TLL (Low) | Ref. | 2.23$^{NS}$ | 3.19$^{NS}$ | 7.02* | Null | Null |
| Plant_CRO | CRO (low) | Ref. | [b]64.34* | 70.77* | 486.43** | Ref. | [c]0.12*** |
| | CRO(middle) | Ref. | ×$^{NS}$ | 2.32$^{NS}$ | 22.49* | Null | Null |
| | CRO (high) | Ref | Null | Null | Null | Null | Null |
| | | Odds ratio of all sediment runoff events | | | | | |
| Soil_SHC | SHC (low) | Ref. | ×$^{NS}$ | 1.22$^{NS}$ | 1.82$^{NS}$ | Null | Null |
| Plant_TLL | TLL (Low) | Ref. | ×$^{NS}$ | 1.22$^{NS}$ | 1.82$^{NS}$ | Null | Null |
| Plant_CRO | CRO (low) | Ref. | ×$^{NS}$ | ×$^{NS}$ | ×$^{NS}$ | Ref. | 0.33** |
| | CRO(middle) | Ref. | ×$^{NS}$ | ×$^{NS}$ | ×$^{NS}$ | Null | Null |
| | CRO (high) | Ref | Null | Null | Null | Null | Null |

a: making the interactive effect of low-grade of SHC and low-grade of ASM as reference, the odds ratio of all random runoff events affected by the interactive effect of low-grade of SHC and middle-grade of ASM is 2.23 times larger than that interactive effect of low-grade SHC and low-grade of ASM under controlled rainfall conditions; b: making the interactive effect of low-grade of CRO and low-grade of ASM as reference, the odds ratio of all random runoff events affected by the interactive effect of low-grade of CRO and middle-grade of ASM is 64.34 times significantly larger than that interactive effect of low-grade of CRO and low-grade of ASM under controlled rainfall conditions, with P≤0.1; c: making the interactive effect of low-grade of CRO and low-grade of TLL as reference, the odds ratio of all random runoff events affected by the interactive effect of low-grade of CRO and high-grade of TLL is 0.12 times significantly larger than that interactive effect of low-grade of CRO and low-grade of TLL, with P≤0.001

(Wald test statistic is applied to test the significant of odds ratio *** P≤0.001, ** P≤0.01, * P≤0.1, NS: not significant, ×$^{NS}$: the nonsignificant value cannot be estimated)


Figures

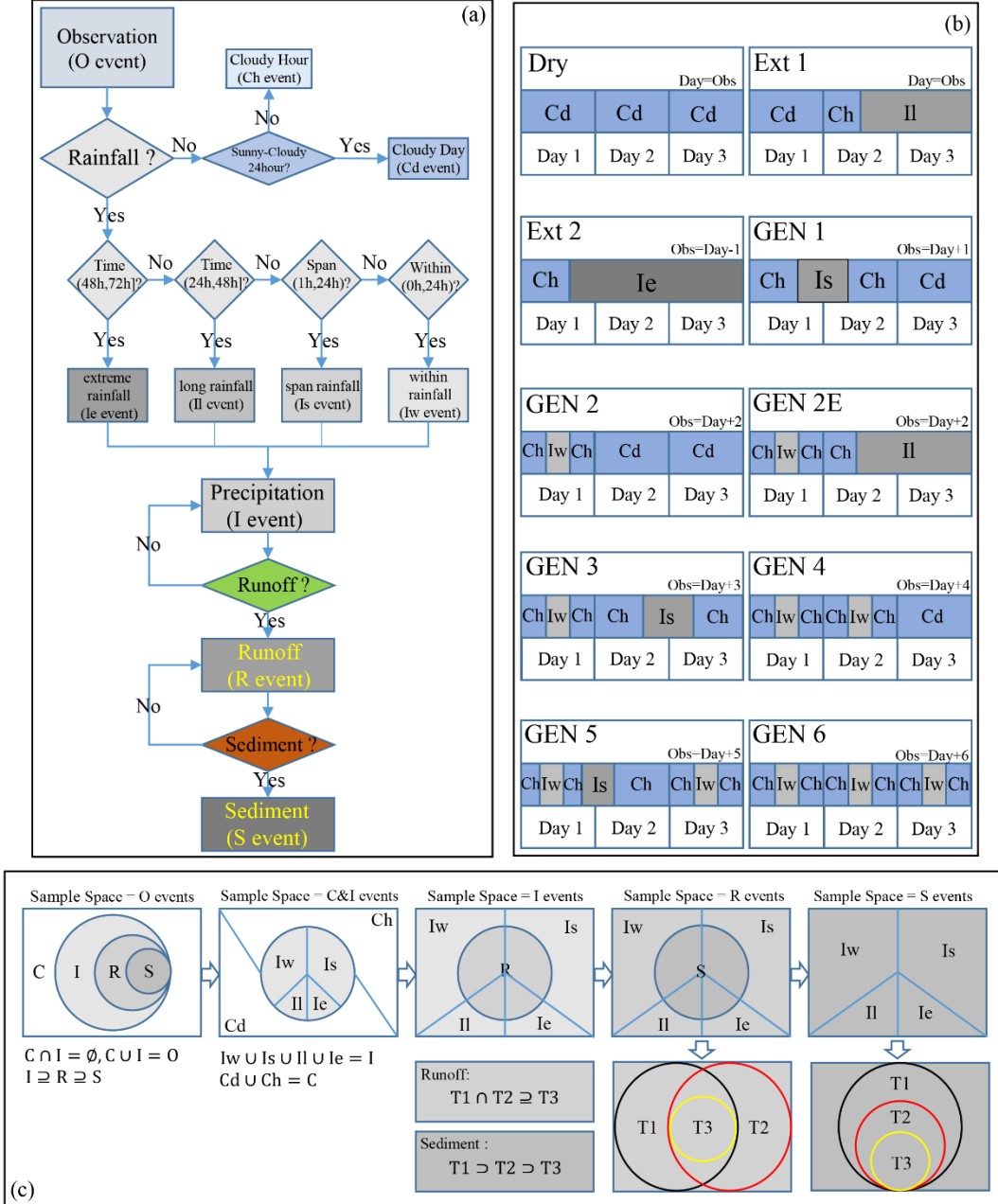

Figure 1    The construction of OCIRS system : (a) a flow chart to determine all random event types in OCIRS framework; (b) the different combining patterns of rainfall and non-rainfall events in three consecutive days to form ten observed random event sequences on five rainy seasons; (c) Venn diagram to reveal the relationship among all random events types in OCIRS framework.







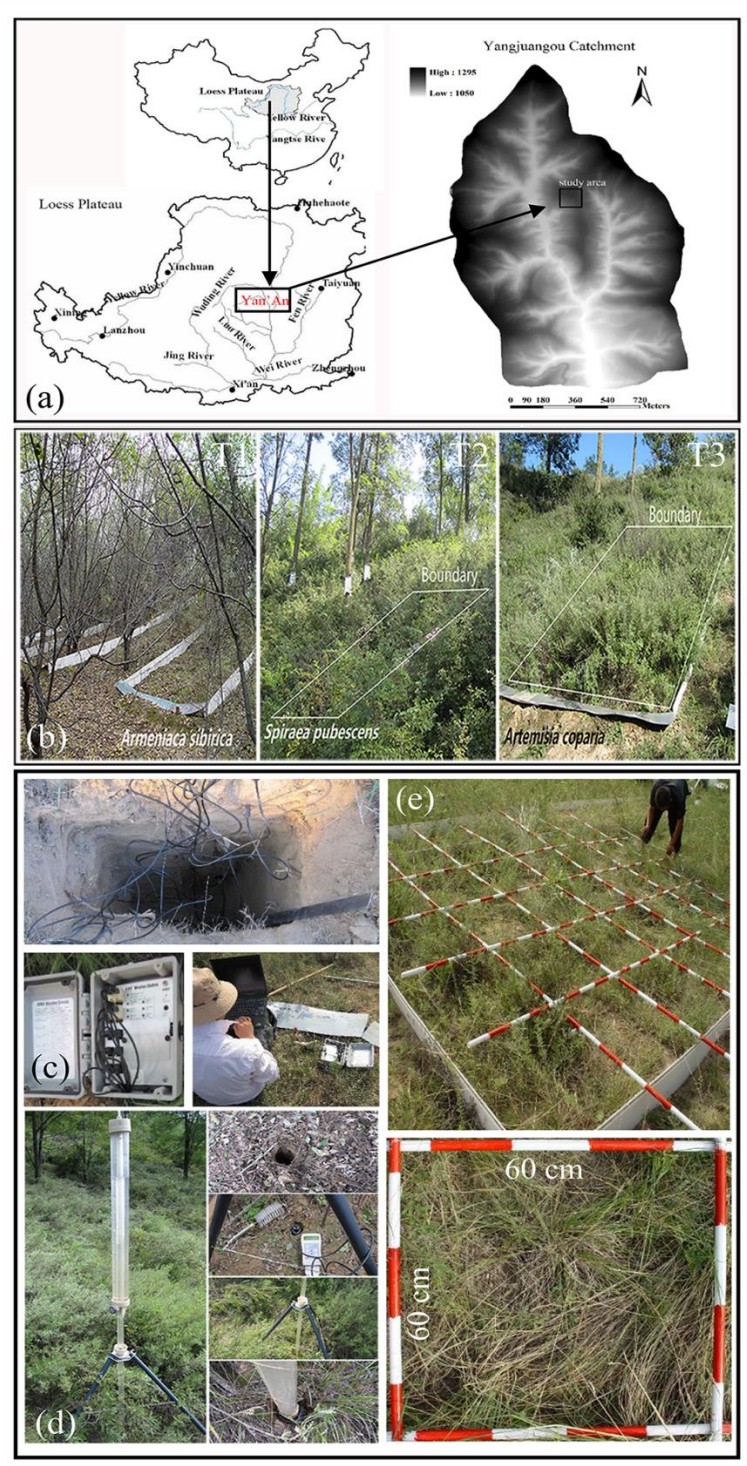

Figure 2    Study area and experimental design: (a) location of the Yangjuangou Catchment; (b) three restoration vegetation types including *Armeniaca sibirica* (T1), *Spiraea pubescens* (T2), and *Artemisia copria* (T3); (c) the dynamic measurement of soil moisture and data collection to provide the information about average antecedent soil moisture; (d) the measurement of field saturated hydraulic conductivity to determine the average infiltration capability; (e): the investigation of morphological properties of restoration vegetation by setting quadrats


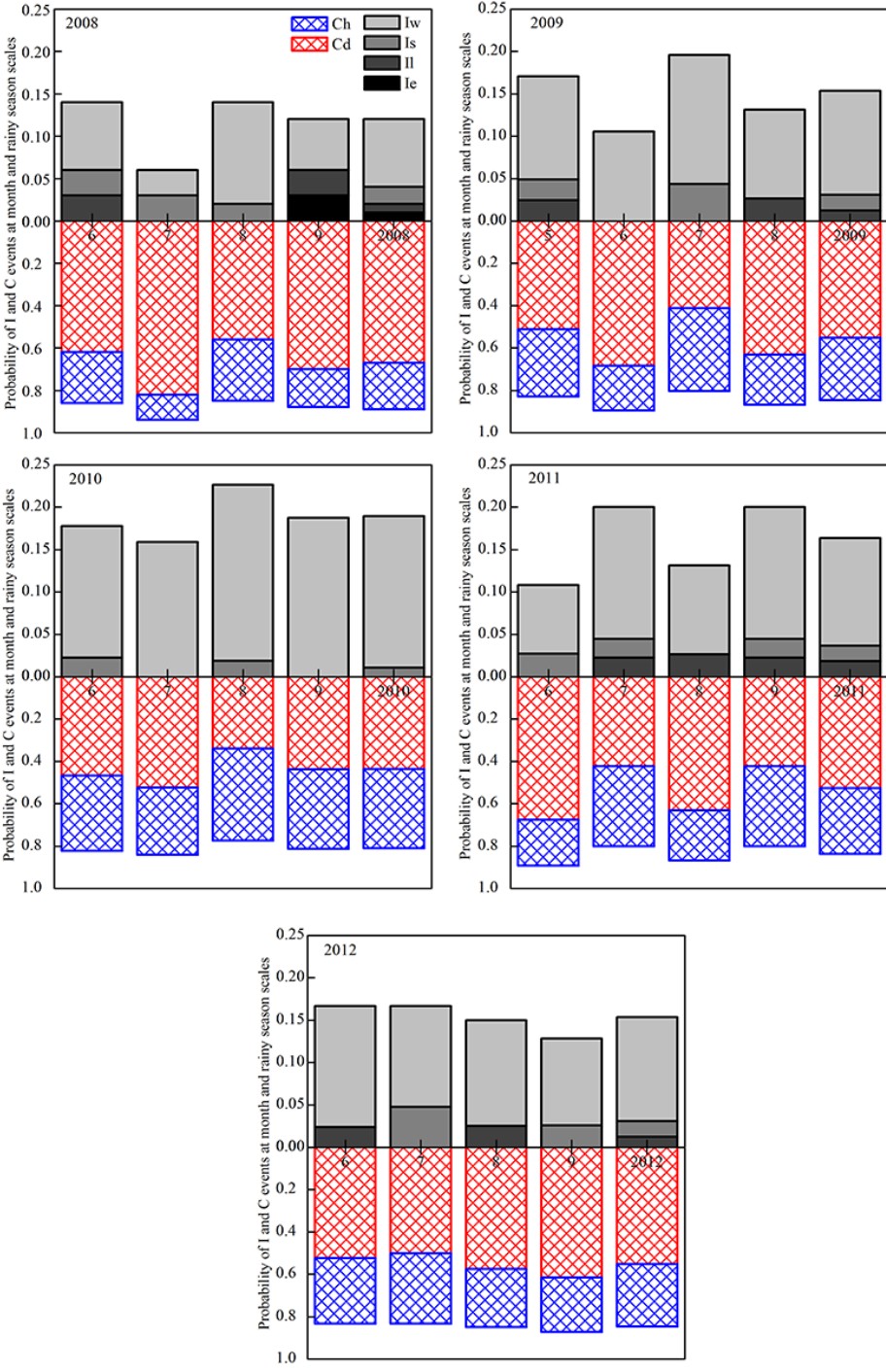

Figure 3    The probability distribution of different random rainfall event types (Iw, Is, Il, and Ie) and random non-rainfall event types (Ch and Cd) at monthly and seasonal scales from rainy season of 2008 to 2012.






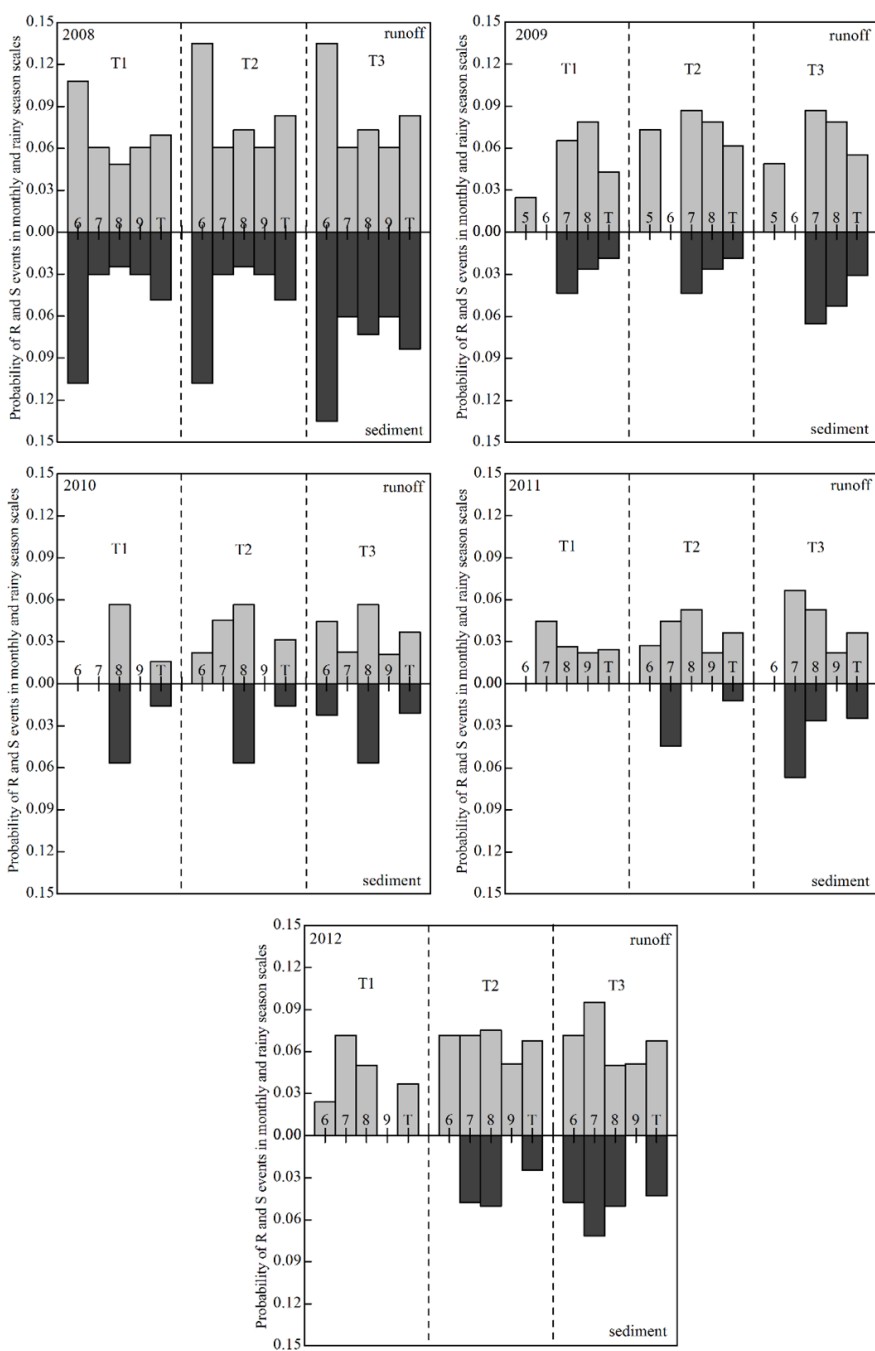

Figure 4     The probability distribution of random runoff and sediment events generating in three restoration vegetation types at monthly and seasonal scales from rainy season of 2008 to 2012, the Arabic numbers and letter "T" on the abscissa indicate the month and season respectively, the same as follow figures







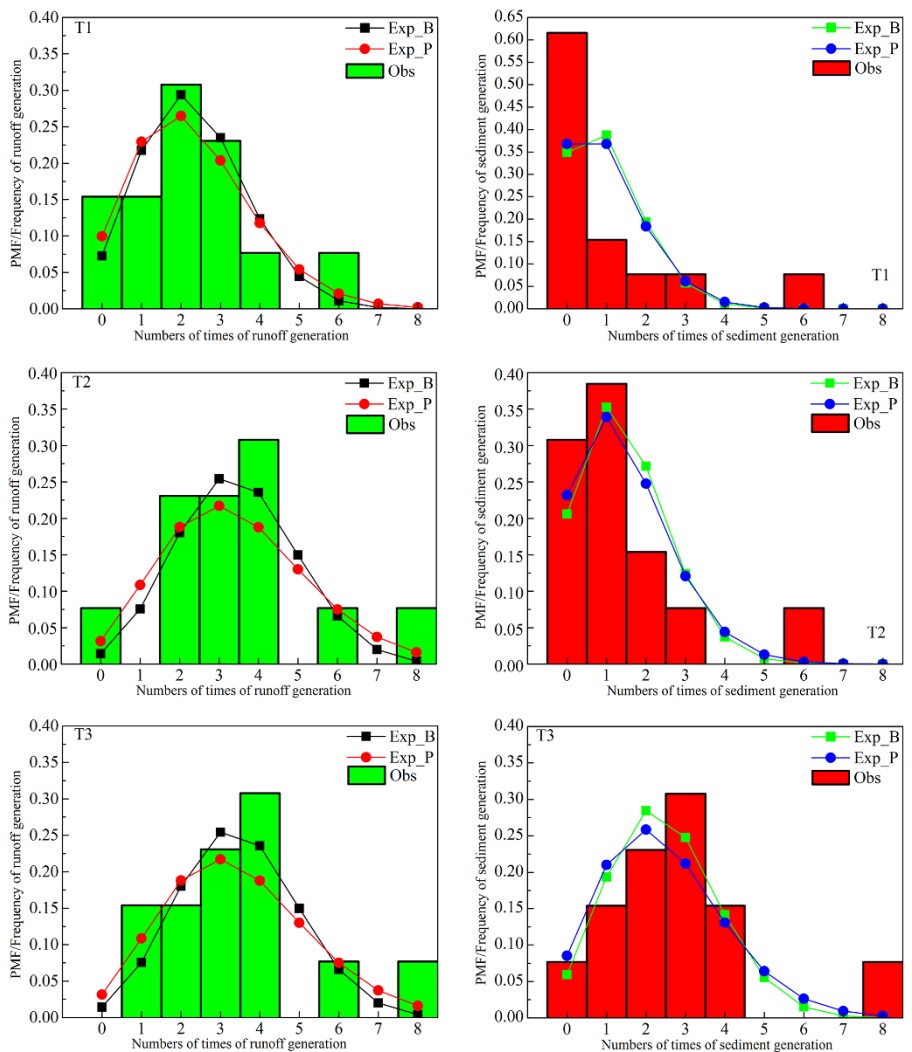

Figure 5     The comparison between simulation of stochasticity of runoff and sediment events by Binomial and Poisson PMFs and the observed frequencies of numbers of times of soil erosion events in three restoration vegetation type, Exp_B and Exp_P indicates the simulated values in Binomial and Poisson PMF respectively, and the histogram represents the observed values.


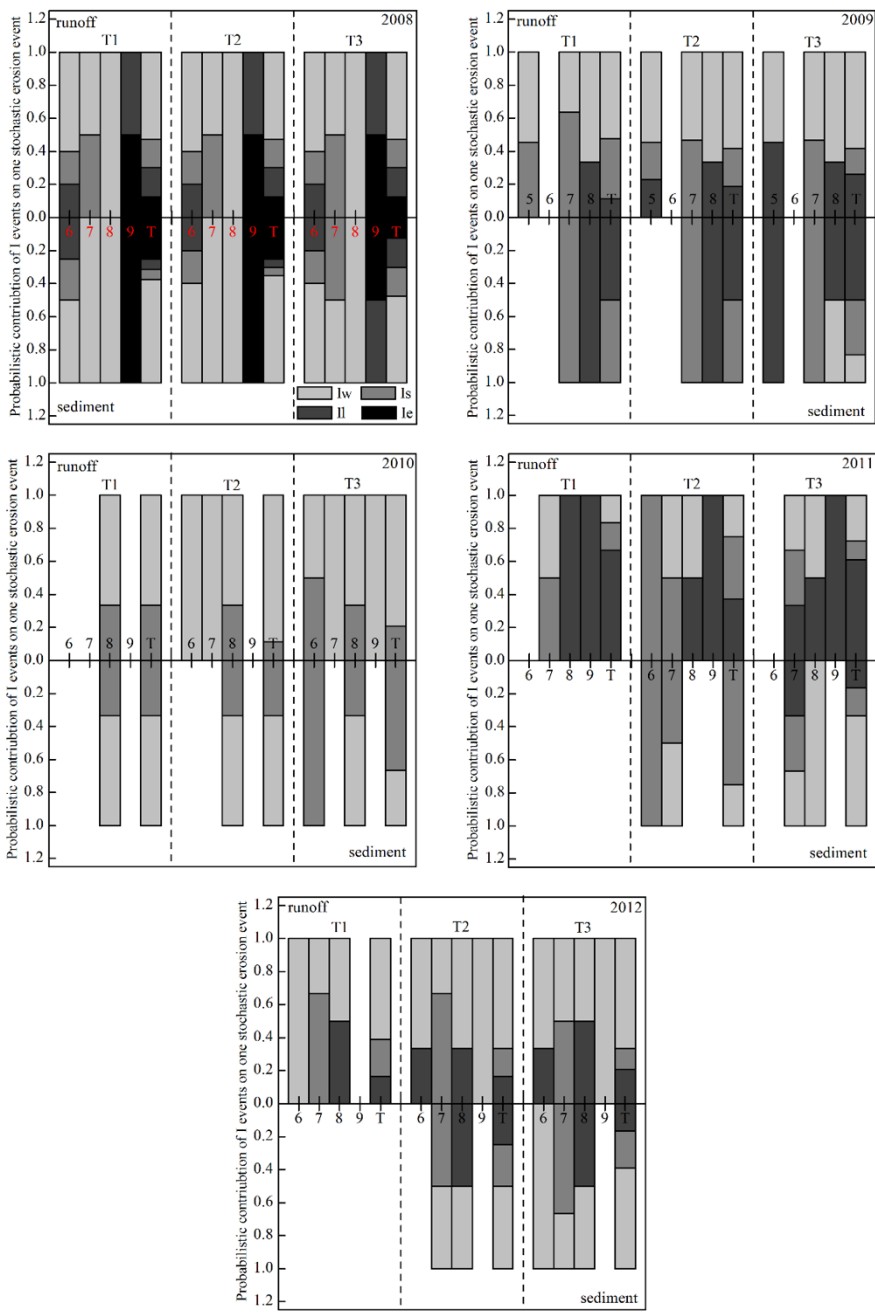

Figure 6     The distribution of probabilistic contribution of four random rainfall event types on anyone runoff or sediment event stochastically generating in three restoration vegetation types at monthly and seasonal scales from rainy season of 2008 to 2012








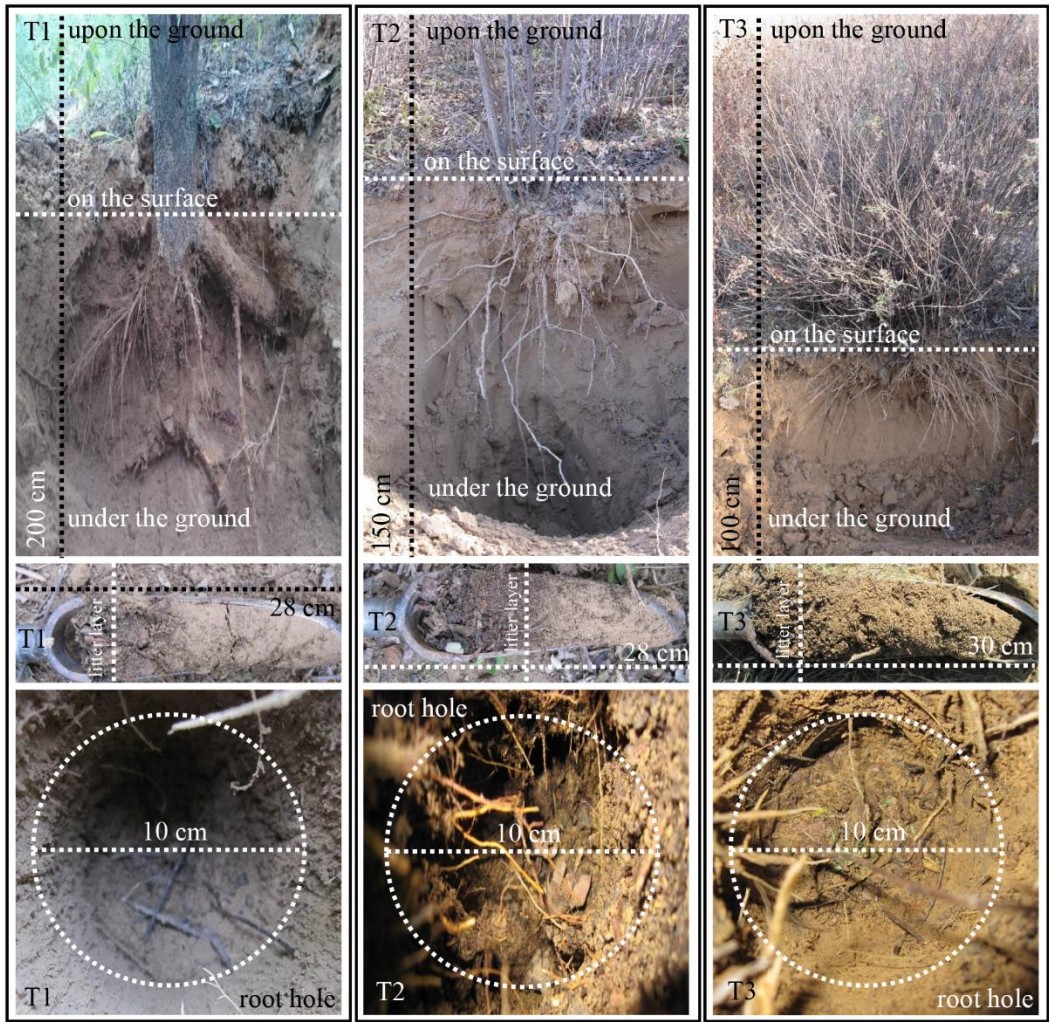

Figure 7      Morphological properties of three restoration vegetation types including the thickness of litter layer, the distribution of root system. The dashed lines indicates the diameter and depth of soil samples with approximating 10 cm and 30 cm respectively.
















**Stage I:  Construction & Determination**

**Step 1:  Constructing OCIRS system**
Collecting and classifying influencing factors to characterize the stochastic environment

**Step 2:  Determing monitoring period**
From short-term to long-term monitoring of erosion events generation.

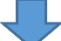

**Stage II:  Observation & Simulation**

**Step 3:  Phased adjustment of description**
*Short-term*: OCIRS-Bayes to analyze stochastic erosion events
*Mid-term*: OCIRS-Binomial to analyze stochastic erosion events
*Long-term*: OCIRS-Poisson to analyze stochastic erosion events

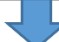

**Stage III: Evaluation & Management**

**Step 4: Probabilistic attribution evaluation**
LRM to determine vegetation types with stronger capacity for reducing probability of erosion generation

**Step 5: Restoration vegetation selection**
Managing to select the restoration vegetation by IPA to improve soil and water conservation

Figure 8     The framework of integrated probabilistic assessment for soil erosion monitoring and restoration strategies
