# Peer review of "An integrated probabilistic assessment to analyze stochasticity of soil erosion in different restoration vegetation"

_Hydrology and Earth System Sciences, 2016_

## Referee Comment (RC1) · Anonymous Referee #1 · 23 Aug 2016

1. General comments: This manuscript integrated the probabilistic theory and experimental data to systematically assess the stochasticity of soil erosion in different restoration vegetation types. It constructed the OCIRS-Bayes analysis framework to describe the stochasticity of various random events in observed environmental conditions, employed probabilistic approaches to quantify and predict the randomness of soil erosion, and finally used statistic inference theories to discuss the prediction effects of different probabilistic models.

It is a very interesting and attractive study, in particular, from the stochasticity perspectives, it probably supplemented a previous soil erosion studies. However, the language and grammar in this text must be carefully checked to improve the readability of this paper, and other some specific comments or flaws of this manuscript are showed as follows:

[Figure]

2. Specific comments or flaws: 2.1 In the abstract section Line 32: the change the "erosion random events" into "random erosion events"

2.2 In the introduction section Lines 93-95: This sentence may be deleted, because this paragraph mainly focus on the previous study and missing information about the probabilistic-trait approach application. Lines 93-95 indicated the value of the application of probabilistic-trait on assessing erosion stochasticity, which should put them on the end of the introduction section.

Lines 111-115: These sentences should also be deleted in this paragraph, or be put on the end of the introduction section, because, in the fourth paragraph, the author highlighted the previous research on the effect of vegetation on soil erosion, as well as pointed out the missing information about the effect of vegetation on erosion stochasticity. Lines 111-115 mainly expressed the value of the assessing the effect of vegetation on erosion stochasticity, which should also be finally concluded in the last part of the introduction section.

Line 117-126: This part should be rewrote or restructured. Because, based on the description of missing information, the author should highlight the statement of the purpose and value of this study. I suggest author should put lines 93-95 and lines111-115 together to add into this part, which probably could make the structure of introduction be more logic and clear.

2.3 In the materials and method section In the section 2.3.1 (Lines 178-206), please modified the terminology about the random event expression, in this section the expression of rainfall random event should modified as random rainfall event, which could more clearly express the meaning of those random variables, meanwhile, please carefully check other terminologies involving the probability theory, such as framework, system, to improve the readability of this paper.

2.4 In the result section In the section 3.2 (Lines 314-351), author used two probabilistic approach to analyze the erosive stochasticity in three restoration vegetation types, why

did you use this two approaches? What is difference between the two approaches? In particular, the application of Bayes formula on stochasticity analysis was seemed to supply more stochastic information about the soil erosion in different restoration vegetation types, what is the meaning of this exploration?

In the section 3.3 (Lines 356-380), author also used two probabilistic distributions to predict the frequencies of the random runoff and sediment events. Why did you use binomial and Poisson distribution to predict the erosion stochasticity instead of choosing other probabilistic distribution functions?

2.5 In the discussion section In the section, author made a specific explanation of the results, discussed about the difference between binomial and Poisson distribution on predicting the stochasticity of soil erosion by using systematic mathematic proving and statistic inference, furtherly highlighting the value of this study. However, in the section 4.1 (Lines 385-422) author should more clearly pointed out the reason for designing the OCIRS-Bayes framework as well as its effect on the stochastic assessment of soil erosion.

2.6 Figures and tables In the figure 4, (Line 944) total reason? Clerical error. In the table 2 and 3 (Line 847-879) contain some repeated information such as the properties of rainfall events, please check up and simplify the two tables.

---

## Referee Comment (RC2) · J.P Puigdefábregas (Referee) · 3 Nov 2016

1. The manuscript shows a novel and interesting approach to link rainfall stochasticity to runoff and sediment delivery stochasticity through the role of vegetation.

2. There is a conceptual inconsistency that the authors should clarify: They claim about the impact of stochasticity on the increase of runoff and erosion, but their measure of stochasticity is a measure of the probability of extreme values or of the classes of frequency values. By this way they ignore memory of the system, which lacking characterizes true stochasticity. This could be obtained through autocorrelation or variograms, which should lack of upper asymptote and therefore the attribute variance grows unbound with the lag or time.

3. The approach lacks explicitly of any theory and totally relies on empirical ad hoc

information from small plots. At the same time, the method is designed to be used in restorations, and as such it should deal with different vegetation, topographies and soil attributes. The number of small plots to grant significance would increase exponentially and so the cost of the operation. How to deal with this issue should be commented by the authors.

4. The parameters used in the transfer probability functions comes from the plots, where the application is performed. This seems incurring in circularity. The authors should clarify that in the interpretationof results.

5. The authors don't mention the spatial stochasticity of rainfall and of the land attributes. A fact crucial in most of large applications supposed of to be real targets of the proposed method 6. The references almost lack mentioning the efforts done since the eighties in the same direction by combining temporal and spatial stochasticity.

---

## Author Comment (AC1) · 9 Nov 2016

We very thank for the referee's some positive comments on our original manuscript, and also appreciate these detailed comments which can undoubtedly improve the quality of this paper. We have carefully read all the comments and make some brief responses by point to point. After that, we will rewrite and restructure the original manuscript on the basis of the referee's useful suggestions and comments.

Comment 1: In the abstract L32. Change the "erosion random events" into "random erosion events"

Response 1: In the original manuscript, the expression of "erosion random events" is incorrect. Random erosion events is right expression stressing the stochasticity of erosion events. We will carefully check the similar incorrect expression in manuscript,

such as changing the rainfall random events into random rainfall events and so on.

Comment 2: In the introduction section L93-95. Deleting this sentence or putting it on the end of introduction

Response 2: In the introduction L93-95 highlights the value for improving the understanding of the stochastic interaction between rainfall and erosion. Just like the referee's comment, the content of L93-95 is irrelevant to the corresponding paragraph. In the revised manuscript, we will delete this sentence and restructure this paragraph.

Comment 3: At the end of introduction (L117-126). Restructuring this part, or adding L93-95 and L111-115 into this part.

Response 3: According to referee's comment, we will adjust this part at end of introduction which claim the aim and meaning of this study in the revised version. In fact, we will also restructure the whole introduction section and make the argument processes to be more logic and clear.

Comment 4: In the method and material section (L178-206). Modification of the terminology about random event expression.

Response 4: There are some incorrected terminological expressions about random events from L178 to L206 which is the similar to the L32 in the abstract. According to the referee's comment, we will carefully check these incorrect expression and modify in revised manuscript.

Comment 5: In the result section (L314-351). Explanation of the reason for using two probabilistic approaches, and the difference between the two approaches

Response 5: In the result section, in order to quantitatively describe the stochasticity of environment affecting the generation of runoff and sediment. Firstly, we introduce the OCIRS system to calculate the probability of random runoff and sediment events occurring in different plot types based on probability theory in the study area. Actually, the OCIRS system could be regarded as an event-driven conceptual model indicating

the relationship between all observed different weather conditions and erosion events based on the exploration of stochastic information. Because, OCIRS provide a platform to compare the risk of erosion generation through probability values. Therefore the causal effects of erosive rainfall events on the randomness of runoff and sediment could be constructed on the basis of the application of OCIRS system.

On the other hand, as a second method, Bayes model, however, could be regard as an "inverse" application of OCIRS on describing the stochastic relationship between environment and erosion events. Because, in this study, the Bayes model could be considered as a feedback of a random erosion events on four different rainfall event types. Just like the referee's mention, the Bayes model supply more stochastic information about erosion properties, it implies how much contribution of rainfall events to any random erosion. Especially, under information deficiency conditions, Bayes model is an important supplement for assessing the randomness of erosive events occurring in different vegetation types.

Consequently, the combination of OCIRS and Bayes model finally form a whole analysis loop patten (OCRIS-Bayes framework) to systematically describe the effect of restoration vegetation on erosion randomness. We will follow the referee's suggestion, and highlight the meaning of OCIRS-Bayes in discussion section.

Comment 6: In the result section (L356-380). Explanation of the reason for using binomial and Poisson distribution function, rather than using other probabilistic distribution functions?

Response 6: In the study, we generally hypothesize that the stochastic information or signal of rainfall is one of most important and indispensable factor to be transmitted into erosion phenomenon and at last affect the randomness of erosion events. Reported by former literatures, especially depending on the relevant research by Eagleson, binomial and Poisson distribution method were applied to describe the probabilistic distribution of rainfall events, which moreover have good predictive effect on annual rainfall events.

Therefore, the closed causal relationship between stochasticity of rainfall and erosion generation is our first reason for selecting binomial and Poisson distribution to describe and predict the probability distribution of runoff and sediment events.

The second reason for choosing binomial and Poisson is that the phenomenon of soil erosion can be simplified to a series random variable satisfying the theoretical hypothesis of binomial distribution. These characteristics of random variables also more satisfy with the fundamental and premise of binomial distribution application than other probabilistic models. As to the response of the referee's comment, we will added the relevant explanation in revised manuscript.

Comment 7: In the discussion section (L385-422). Making clearer explanation of the reason for designing OCIRS-Bayes framework

Response 7: This comment is similar with comment 5. We will follow the referee's suggestion in revised version to highlight the reason and meaning of introducing and employing the OCIRS-Bayes framework on the stochasticity of soil erosion.

Comment 8 Checking the clerical error in figure caption and simplifying the content of tables

Comment 8 The "total reason" in L944 is an obvious clerical error, we are very sorry, and we will change it to "total season or whole season". We will also adjust the content in table 2 and 3 to make the information in these two tables be more effectively indicate the main properties of different random rainfall event types. Finally, we will carefully check other clerical errors in original manuscript and probably invite a native English speaker to polish the language.
* * *

---

## Author Comment (AC2) · 28 Nov 2016

Dear Prof. Puigdefábregas:

We very thank for your suggestion to our manuscript. Your comments and suggestion give us great inspiration and help to improve the quality of this paper greatly.

We also appreciate and admire the accomplishments you and your colleagues have achieved in the soil erosion science, and specially, the vegetation-driven spatial heterogeneity (VDSH) theory proposed by you in 2005, give us deep impression for studying the relationship between soil erosion and vegetation patterns. Because we believe this theory provides a new perspective for exploring the role of vegetation acting on the erosion processes in water-limited environment. Moreover, some of your other studies conducting in Spain also enlighten our study focusing on the soil erosion in the Loess

[Figure]

Plateau. It is a great honor for us to get your guidance and suggestion for our erosion study.

We have carefully read all the comments and suggestions, and also have gathered together to discussed some of suggestions very carefully. According to your and another anonymous referee suggestions, we will rewrite and restructure the original manuscript. At first, we make some brief responses by point to point to your suggestion and comments as follows:

Comment 1: Their measure of stochasticity is a measure of probability of extreme values or of the classes of frequency values, and ignore memory of the system, which lacking characterizes true stochasticity.

Response 1: In the original manuscript, the probability of soil erosion was measured by the frequency values of runoff and sediment events generating over five rainy seasons depending on the observational data.

The frequency value could be regarded as some of properties of erosion stochasticity, because all the erosion events were triggered by stochastic rainfall events, and to some extent, the generation of soil erosion could be regarded as a result of how the random signals of rainfall to be transmitted into the soil system and finally generate erosion events.

Prof. Puigdefábregas mentioned the ignorance of memory of the system in this paper, which gave us a very important suggestion to improve our original manuscript.

According to our field observation, we believed that, besides the randomness of rainfall events, the properties of plants and soil could also be the main factors to impact on the probability of soil erosion, and further affect the memory of the system, therefore, we will modify the original manuscript from the following aspects: 1. Make clear clarification of the stochasticity of soil erosion in the revised manuscript. 2. Reclassify and redefined all the observed rainfall events types to highlight their roles playing on the

quantifying the probability of soil erosion in the revised manuscript. (more explanation is in supplementary ) 3. Highlight the effects of the properties of plant and soil on the randomness of runoff and sediment events generating in different vegetation types.

Comment 2: The approach lacks explicit of any theory and totally relies on empirical ad hoc information from small plots. And the method is designed to be used in restorations, and as such it should deal with different vegetation, topographies and soil attributes. How to deal with this issue should be commented by the authors

Response 2: In the revised manuscript, we will introduce the logistic regression method to analyze the effect of vegetation and soil hydrological properties on the probability of soil erosion.

In the method section of revised manuscript, we will highlight that why the theory of binomial and Poisson distribution function could be used to describe the randomness of soil erosion, and what the difference is between binomial and Poisson distribution applied on the calculation of erosion stochasticity.

Actually, in this paper nearly all the empirical ad hoc information from small plots were quantified by the probability theories from Bayes theories to binomial-Poisson theories as well as to a series of point estimation theories.

In revised manuscript, we will hope to explore a method to systematically describe the probability of soil erosion by using Binomial-Poisson method, as well as to make attribution-analysis of randomness of erosion phenomenon by using Bayes and logistic regression method. Consequently, the combination of probability theories and model could form an integrated exploring framework to analyze the erosion randomness in different vegetation types.

Secondly, we admitted the limitation of the experiment design in the study. Just as Prof. Puigdefábregas' mention, the increasing of vegetation, topographies and soil attributes will increase the numbers of small plots as well as increase the cost of operation,

we will comment our experimental limitation in the discussion section of the revised manuscript.

According to Prof. Puigdefábregas' suggestion, in the next step of erosion stochasticity study, we will try to construct some bare plots in the study area to collected more random information of soil erosion to enrich the understanding of stochastic property of erosion in different land covers.

Comment 3: The parameter used in the transfer probability functions comes from the plots, where the application is performed. This seems incurring in circularity. The authors should clarify that in the interpretation of results.

Response 3: The circularity of argument could probably related to our unclear expression in paper. When we received this comment of Prof. Puigdefábregas, we came together and carefully discussed the meaning of application of binomial and Poisson distribution function in original manuscript, and finally concluded that:

1.The application of binomial and Poisson probability function could act as an important role on detailing the stochastic information of soil erosion in different restoration vegetation types under month scale, rather than on predicting randomness of soil erosion mentioned by THE original manuscript. Therefore in the revised manuscript, we will modified former expression.

2.The purpose of application binomial and Poisson probability function is to select more appropriate method to describe the stochastic property of erosion in detail. According to the point estimation depending on the maximum likelihood estimator and uniformly minimum variance unbiased estimator, Poisson probability function was found to be more appropriate for describing the probability of erosion generation in long-term monitoring period. Consequently, we re-establish the whole logical structure in revised manuscript as follows:

(1)Proposing hypothesis: Randomness of soil erosion is one of important properties

of erosion phenomenon, how to systematically describe the stochasticity of erosion depending on long-term field observations? And how the rainfall, vegetation and soil properties affects the stochasticity of erosion?

(2)Testing hypothesis: First, take the conditional probability to describe the probability of runoff and sediment events under rainy season scales; secondly, apply binomial and Poisson probability function to describe the randomness of soil erosion in detail on month scale, and compare the observed frequency distribution with probability distribution; Thirdly, analyze the effect of properties of rainfall, vegetation and soil saturated hydraulic conductivity on the random runoff and sediment events by using logistic regression models; finally propose that the multiple-probability models could be regarded as an integrated probabilistic assessment to analyze stochasticity of soil erosion.

(3)Discussing hypothesis: First, make the parameter estimation to compare the appropriative of application of binomial and Poisson probability distribution on stochascity description. Secondly, explain the role of vegetation and soil properties acting on affecting the probability of soil erosion in different restoration vegetation types. Thirdly, mention the meaning and application of the integrated probabilistic assessment on soil erosion study, and the limitation of the experiment design. Consequently, the adjusted logical structure in revised manuscript may be avoid the circularity in whole argument processes.

Comment 4: The author don not mention the spatial stochasticity of rainfall and of the land attributes. The references almost lack mentioning the efforts done since the eighties in the same direction by combining temporal and spatial stochasticity.

Response 4: Thanks for Prof. Puigdefábregas' suggestion. We will supplement the contribution and efforts of temporal and spatial stochasticity in introduction section of revised manuscript. As Prof. Puigdefábregas' mention, there exist spatial stochasticity of rainfall and of the land attributes, however, we mainly focused on the plot scale, and to same extent, assume precipitation and soil characteristics in plot scale are continuous. The properties of different soil saturated hydraulic conductivity in the three vegetation types could probably affect the stochasticity of soil erosion, which will be discussed by using logistic regression method in the revised manuscript.

Finally, we thank again for Prof. Puigdefábregas' great help and guidance for improving our study on soil erosion.

Please also note the supplement to this comment:
http://www.hydrol-earth-syst-sci-discuss.net/hess-2016-386/hess-2016-386-AC2-supplement.pdf

**Supplement:**

**Supplement:   Description frameworks of random events**

Each observed stochastic weather condition with different durations in field monitoring period was defined as a random experiment. All possible outcomes of a random experiment constituted a sample space ($\Omega$) defined as a random observational event (short for O event). Two mutually exclusive random event types—random rainfall event (I event) and random non-rainfall event (C event)—constituted the O event. Precipitation is a necessary condition of runoff production, therefore, the random runoff production event (R event) is a subset of I event. Similarly, R event is also a necessary condition of random sediment migration event (S event), causing S is contained in R. As a result, O, C, I, R, and S events constituted a random events framework (OCIRS) to describe the stochasticity of environment.

The random event duration in OCIRS is an important property of stochastic weather conditions. In particular, the duration property of I event was closely related to the transmission of stochastic signals of rainfall into the R and S events. According to the rainfall duration patterns in China (Wei et al., 2007; Yin et al., 2014), the time interval between two adjacent individual I events is set to be more than 6 hours, forming the criteria for individual rainfall classification. Therefore, we summarized duration property of all I events and classified them into four mutually exclusive I event types. They were a random extreme long rainfall event type (Ie event), a random general long duration rainfall event type (Il event), a random spanning rainfall event type (Is event) whose duration spans two consecutive days, and a random within rainfall event type (Iw event) generated in a day. Additionally, the C event can also be divided into two

types at daily scale. They are random non-rainfall event type lasting a whole day (Cd event), and random non-rainfall event type whose duration is less than 24 hours (Ch event) which is interrupted by I event. Table 1 summarized the physical, probabilistic properties and implication of all random event types in OCIRS. The determining process of all random event types in OCIRS was sketched by Figure 1a, and Venn diagrams in Figure 1c explored the relationships of all random event types in OCIRS.

In fact, various combinations of I and C events formed different random event sequences, constituting the observed stochastic weather condition over field monitoring period. Considering the observed longest duration of Ie event approximating 72 hours (Table 1), we defined a random event sequence unit (RESU) as a combing pattern of I and C events in three consecutive days, and summarized ten observed RESUs in five rainy seasons (Figure 1b).

[Figure]

**Figure 1.** The OCIRS-RESU system: (a) a flow chart to determine all random event types in OCIRS framework; (b) the different combining patterns of rainfall and non-rainfall events in three consecutive days to form ten observed RESUs on five rainy seasons; (c) Venn diagram to reveal the relationship among all random events types in OCIRS framework.

**Table 1.** Physical and probabilistic meanings of all random event types in OCIRS

[revised manuscript text omitted]

---

## Author Response (AR1)

[revised manuscript text omitted]

**Authors' Response:**

**Response to Anonymous Referee #1**

We very thank for the referee's positive comments on our original manuscript, and also appreciate these detailed comments which can undoubtedly improve the quality of this paper. We have carefully read all the comments and make some brief responses by point to point. In the revised manuscript, we have rewritten and restructured the original manuscript on the basis of the referee's useful suggestions and comments.

**Comment 1:**
In the abstract L32. Change the "erosion random events" into "random erosion events"

Response 1:
In the revised manuscript, we have changed the incorrect expression. We also have carefully checked the similar incorrect expression.

**Comment 2:**
In the introduction section L93-95. Deleting this sentence or putting it on the end of introduction

Response 2:
We have rewritten the introduction section in the revised manuscript, and have deleted this sentence.

**Comment 3:**
At the end of introduction (L117-126). Restructuring this part, or adding L93-95 and L111-115 into this part.

Response 3:
We have adjusted this part and rewritten the aim and meaning of this study in the line 168-181 in the introduction section of revised manuscript.

**Comment 4:**
In the method and material section (L178-206). Modification of the terminology about random event expression.

Response 4:
There are some incorrected terminological expressions about random events from L178 to L206. We have rewritten and supplemented this part in revised manuscript from line 187 to line 219.

**Comment 5:**
In the result section (L314-351). Explanation of the reason for using two probabilistic approaches, and the difference between the two approaches

Response 5:
In the result section, in order to quantitatively describe the stochasticity of environment affecting the generation of runoff and sediment. Firstly, we introduce the OCIRS system to calculate the probability of random runoff and sediment events occurring in different plot types based on probability theory in the study area. Actually, the OCIRS system could be regarded as an event-driven conceptual model indicating the relationship between all observed different weather conditions and erosion events based on the exploration of stochastic information. Because, OCIRS provide a platform to compare the risk of erosion generation through probability values. Therefore the causal effects of erosive rainfall events on the randomness of runoff and sediment could be constructed on the basis of the application of OCIRS system.

On the other hand, as a second method, Bayes model could be regard as an "inverse" application of OCIRS on describing the stochastic relationship between environment and erosion events. Because, in this study, the Bayes model could be considered as a feedback of a random erosion events on four different rainfall event types. Just like the referee's mention, the Bayes model supply more stochastic information about erosion properties, it implies how much contribution of rainfall events to any random erosion. Especially, under information deficiency conditions, Bayes model is an important supplement for assessing the randomness of erosive events occurring in different vegetation types.

We have follow the referee's suggestion, and systematical discussed the meaning and implication of the two probabilistic models in the line 527-597 in the revised manuscript.

**Comment 6:**
In the result section (L356-380). Explanation of the reason for using binomial and Poisson distribution function, rather than using other probabilistic distribution functions?

Response 6:
In the study, we generally hypothesize that the stochastic information or signal of rainfall is one of most important and indispensable factor to be transmitted into erosion phenomenon. Reported by former literatures, especially depending on the relevant research by Eagleson, Binomial and Poisson distribution method were applied to describe the probabilistic distribution of rainfall events, which moreover have good predictive effect on annual rainfall events. Therefore, the closed causal relationship between stochasticity of rainfall and erosion generation is our first reason for selecting binomial and Poisson distribution to describe and predict the probability distribution of runoff and sediment events.

The second reason for choosing binomial and Poisson is that the phenomenon of soil erosion can be simplified to a series random variable satisfying the theoretical hypothesis of binomial distribution. These characteristics of random variables also more satisfy with the fundamental and premise of binomial distribution application than other probabilistic models. As to the response of the referee's comment, we have added the relevant explanation in revised manuscript in discussion section.

**Comment 7:**
In the discussion section (L385-422). Making clearer explanation of the reason for designing OCIRS-Bayes framework

Response 7:
This comment is similar with comment 5 and 6. We have followed the referee's suggestion, and added the interpretation in discussion section of revised manuscript.

**Comment 8**
Checking the clerical error in figure caption and simplifying the content of tables

Response 8
The "total reason" in L944 is an obvious clerical error, we are very sorry, and we have changed it in revised manuscript. We have carefully checked other clerical errors in original manuscript and invited a native English speaker to polish the language.

**Response to Prof. Puigdefábregas' comments and suggestion**

**Dear Prof. Puigdefábregas:**

We very thank for your suggestion to our manuscript. Your comments and suggestion give us great inspiration and help to improve the quality of this paper greatly.

We also appreciate and admire the accomplishments you and your colleagues have achieved in the soil erosion science. Especially, the vegetation-driven-spatial-heterogeneity (VDSH) theory proposed by you in 2005, give us deep impression for studying the relationship between soil erosion and vegetation patterns. Because we believe this theory provides a new perspective for exploring the role of vegetation acting on the erosion processes in water-limited environment. Moreover, some of your other studies conducting in Spain also enlighten our study focusing on the soil erosion in the Loess Plateau. It is a great honor for us to receive your guidance and suggestion for our erosion study.

We have carefully read all the comments and suggestions, and also have gathered together to discussed some of suggestions very carefully. According to your and another anonymous referee suggestions, we have rewritten and restructured the original manuscript. At first, we make some brief responses by point to point to your suggestion and comments as follows:

**Comment 1**:

Their measure of stochasticity is a measure of probability of extreme values or of the classes of frequency values, and ignore memory of the system, which lacking characterizes true stochasticity.

Response 1:

In the original manuscript, the probability of soil erosion was measured by the frequency values of runoff and sediment events generating over five rainy seasons depending on the observational data.

The frequency value could be regarded as some of properties of erosion stochasticity, because all the erosion events were triggered by stochastic rainfall events, and to some extent, the generation of soil erosion could be regarded as a result of how the random signals of rainfall to be transmitted into the soil system and finally generate erosion events.

Prof. Puigdefábregas mentioned the ignorance of memory of the system in this paper, which gave us a very important suggestion to improve our original manuscript.

According to our field observation, we believed that, besides the randomness of rainfall events, the properties of plants and soil could also be the main factors to impact on the probability of soil erosion, and further affect the memory of the system, therefore, we have modified the original manuscript from the following aspects:
1. Make clear clarification of the stochasticity of soil erosion in the revised manuscript.
2. Reclassify and redefined all the observed rainfall events types to highlight their roles playing on the quantifying the probability of soil erosion in the revised manuscript. (more explanation is in supplementary )
3. Highlight the effects of the properties of plant and soil on the randomness of runoff and sediment events generating in different vegetation types by using logistic regression model in the revised manuscript.

**Comment 2:**

The approach lacks explicit of any theory and totally relies on empirical ad hoc information from small plots. And the method is designed to be used in restorations, and as such it should deal with different vegetation, topographies and soil attributes. How to deal with this issue should be commented by the authors

Response 2:

In the revised manuscript, we have supplemented the logistic regression method to analyze the effect of vegetation and soil hydrological properties on the probability of soil erosion.

In the discussion section of revised manuscript, we have highlighted that why the theory of Binomial and Poisson distribution functions could be used to describe the randomness of soil erosion, and what the difference is between binomial and Poisson distribution applied on the calculation of erosion stochasticity.

Actually, in this paper nearly all the empirical ad hoc information from small plots were quantified by the probability theories from Bayes theories to binomial-Poisson theories as well as to a series of point estimation theories.

In revised manuscript, we have tried to explore a method to systematically describe the probability of soil erosion by using Binomial-Poisson method, as well as to make attribution-analysis of randomness of erosion phenomenon by using Bayes and logistic regression method. Consequently, the combination of probability theories and model could form an integrated probabilistic assessment to analyze the erosion randomness in different vegetation types.

Secondly, we admitted the limitation of the experiment design in the study. Just as Prof. Puigdefàbregas' mention, the increasing of vegetation, topographies and soil attributes will increase the numbers of small plots as well as increase the cost of operation, we have commented in the discussion section of the revised manuscript.

According to Prof. Puigdefàbregas' suggestion, in the next step of erosion stochasticity study, we will try to construct some bare plots in the study area to collected more random information of soil erosion to enrich the understanding of stochastic property of erosion in different land covers.

**Comment 3:**
The parameter used in the transfer probability functions comes from the plots, where the application is performed. This seems incurring in circularity. The authors should clarify that in the interpretation of results.

Response 3:
The circularity of argument could probably related to our unclear expression in paper. When we received this comment of Prof. Puigdefàbregas, we came together and carefully discussed the meaning of application of binomial and Poisson distribution function in original manuscript, and finally concluded that:

1. The application of binomial and Poisson probability function could act as an important role on detailing or simulating the stochastic information of soil erosion in different restoration vegetation types under month scale, rather than on predicting randomness of soil erosion mentioned by the original manuscript. Therefore in the revised manuscript, we have modified former expression.

2. The purpose of application binomial and Poisson probability function is to select more appropriate method to describe the stochastic property of erosion in detail. According to the point estimation depending on the maximum likelihood estimator and uniformly minimum variance unbiased estimator, Poisson probability function was found to be more appropriate for describing the probability of erosion generation in long-term monitoring period.

Consequently, we re-establish the whole logical structure in revised manuscript as follows:

(1) Proposing hypothesis: Randomness of soil erosion is one of important properties of erosion phenomenon, how to systematically describe the stochasticity of erosion depending on long-term field observations? And how the rainfall, vegetation and soil properties affects the stochasticity of erosion?

(2) Testing hypothesis: First, take the conditional probability to describe the probability of runoff and sediment events under rainy season scales; secondly, apply binomial and Poisson probability function to simulate the randomness of soil erosion in detail on month scale, and compare the observed frequency distribution with simulated probability distribution; Thirdly, analyze the effect of properties of rainfall, vegetation and soil saturated hydraulic conductivity on the random runoff and sediment events by using logistic regression models; finally propose that the multiple-probability models could be regarded as an integrated probabilistic assessment to analyze stochasticity of soil erosion.

(3) Discussing hypothesis: First, make the parameter estimation to compare the appropriative of application of Binomial and Poisson probability distribution on stochasticity description. Secondly, explain the role of vegetation and soil properties acting on affecting the probability of soil erosion in different restoration vegetation types. Thirdly, mention the meaning and implication of the integrated probabilistic assessment on soil erosion study

Consequently, the adjusted logical structure in revised manuscript may be avoid the circularity in whole argument processes.

**Comment 4:**
The author don not mention the spatial stochasticity of rainfall and of the land attributes. The references almost lack mentioning the efforts done since the eighties in the same direction by combining temporal and spatial stochasticity.

Response 4:
Thanks for Prof. Puigdefàbregas' suggestion. We have supplemented the contribution and efforts of temporal and spatial stochasticity in introduction section of revised manuscript. As Prof. Puigdefàbregas' mention, there exist spatial stochasticity of rainfall and of the land attributes, however, we mainly focused on the plot scale, and to same extent, assume precipitation and soil characteristics in plot scale are continuous. The properties of different soil saturated hydraulic conductivity in the three vegetation types could probably affect the stochasticity of soil erosion, which have discussed by using logistic regression method in the revised manuscript.

Finally, we thank again for Prof. Puigdefàbregas' great help and guidance for improving our study on soil erosion.